https://doi.org/10.1038/s41467-020-18074-8　　**OPEN**

# Interneuron-specific plasticity at parvalbumin and somatostatin inhibitory synapses onto CA1 pyramidal neurons shapes hippocampal output

Matt Udakis[1,3], Victor Pedrosa[2,3], Sophie E. L. Chamberlain[1], Claudia Clopath [2,4✉] & Jack R. Mellor [1,4✉]

The formation and maintenance of spatial representations within hippocampal cell assemblies is strongly dictated by patterns of inhibition from diverse interneuron populations. Although it is known that inhibitory synaptic strength is malleable, induction of long-term plasticity at distinct inhibitory synapses and its regulation of hippocampal network activity is not well understood. Here, we show that inhibitory synapses from parvalbumin and somatostatin expressing interneurons undergo long-term depression and potentiation respectively (PV-iLTD and SST-iLTP) during physiological activity patterns. Both forms of plasticity rely on T-type calcium channel activation to confer synapse specificity but otherwise employ distinct mechanisms. Since parvalbumin and somatostatin interneurons preferentially target perisomatic and distal dendritic regions respectively of CA1 pyramidal cells, PV-iLTD and SST-iLTP coordinate a reprioritisation of excitatory inputs from entorhinal cortex and CA3. Furthermore, circuit-level modelling reveals that PV-iLTD and SST-iLTP cooperate to stabilise place cells while facilitating representation of multiple unique environments within the hippocampal network.

[1] Center for Synaptic Plasticity, School of Physiology, Pharmacology and Neuroscience, University of Bristol, University Walk, Bristol BS8 1TD, UK.
[2] Department of Bioengineering, Imperial College London, London, UK. [3]These authors contributed equally: Matt Udakis, Victor Pedrosa. [4]These authors jointly supervised this work: Claudia Clopath, Jack R. Mellor. ✉email: C.Clopath@Imperial.ac.uk; Jack.Mellor@Bristol.ac.uk

GABAergic inhibitory interneurons form a diverse array of specialised cell types critical for the regulation of complex network functions within the brain. A defining feature of inhibitory interneurons is their precise axonal aborisations whereby inhibitory synapses target specific subdomains of pyramidal neurons and other inhibitory interneurons[1,2]. Within the hippocampus and neocortex, parvalbumin (PV) and somatostatin (SST) expressing interneurons form two broad and occasionally overlapping subtypes of interneurons that preferentially target perisomatic and distal dendritic regions of pyramidal neurons, respectively, and are active on different phases of the theta cycle[2–6]. This endows them with unique roles in sculpting pyramidal neuron responses to excitatory inputs[7,8]. Perisomatic inhibition by PV interneurons regulates pyramidal neuron spiking and network oscillations through feedforward and feedback inhibition[9–12]. In contrast, dendritic inhibition by SST interneurons regulates local synaptic and dendritic conductances, $Ca^{2+}$ signal generation and excitatory synaptic plasticity principally through feedback inhibition[9,13–15].

A defining feature of the hippocampus is the encoding of spatially relevant information via the formation of place cells[16]. Synaptic plasticity at glutamatergic synapses in the hippocampus accounts for the formation of location-specific firing of individual place cells, but it also plays a major role in the formation of place cell assemblies during exploration and offline replay of place cell activity[17–21]. Interestingly, individual CA1 pyramidal neurons can encode distinct place fields in different environments[22] presumably driven by ongoing excitatory synaptic plasticity. This feature of place cells and the persistent plasticity of their synaptic connections present fundamental problems for hippocampal networks balancing flexibility versus stability of representations[23]. It is not clear how place cell assemblies in the hippocampus can encode multiple different locations in separate environments without interference.

Inhibitory interneurons play an integral role within the hippocampus controlling place cell activity[8,24–26], where short-term changes in SST and PV interneuron activity differentially regulate the emergence and firing patterns of place cells[8,25] by controlling glutamatergic synaptic plasticity[13,15]. But the consequences of long-term plasticity at inhibitory synapses on place cell activity have not been investigated.

Long-term inhibitory synaptic plasticity is a potentially important mechanism for learning within cortical networks[27–31] and GABAergic synapses in the hippocampus exhibit structural and functional plasticity[32–34]. Reductions in inhibitory strength via retrograde endocannabinoid signalling is well established[35–37] but multiple other mechanisms to regulate long-term inhibitory synaptic strength have also been proposed including $GABA_B$ receptors and BDNF[38], spike timing-dependent plasticity of chloride transporters[39], retrograde nitric oxide signalling[40] and NMDA receptors[41]. In the neocortex, synapses from PV and SST interneurons can undergo unique forms of plasticity[38,41], whilst in the hippocampus, recent evidence suggests that interneuron subtype-specific inhibitory synapses are regulated in distinct ways[15,42]. However, it is not clear whether long-term plasticity of inhibitory synapses is differentially engaged between interneuron subtypes during physiologically relevant activity and, furthermore, what the consequences of such plasticity would be for hippocampal network activity and place cell representations.

To address these questions, we utilised an optogenetic approach in hippocampal slices to selectively activate perisomatic and dendritically targeting inhibition onto CA1 pyramidal neurons by expression of channelrhodopsin in PV or SST interneurons. We found that synapses from PV and SST interneurons undergo interneuron-specific forms of inhibitory synaptic plasticity driven by the relative timing of inhibitory and excitatory

neuronal spiking and employing distinct signalling mechanisms. We go on to show that these forms of cell-specific long-term inhibitory plasticity have profound effects on the output of CA1 pyramidal neurons and use computational modelling to demonstrate that these plasticity rules can provide a mechanism by which hippocampal place fields can remain stable over time whilst flexibly encoding location in multiple environments.

## Results

**Divergent inhibitory plasticity at PV and SST synapses.** To achieve subtype-specific control of inhibitory interneurons, we selectively activated either PV or SST interneurons by expressing the light-activated cation channel channelrhodopsin-2 (ChR2) in a cre-dependent manner using mice that expressed cre recombinase under control of the promoter for either the PV gene (PV-cre) or SST gene (SST-cre) crossed with mice expressing cre-dependent ChR2 (PV-ChR2 and SST-ChR2 mice; "Methods"). Immunohistochemisty confirmed that ChR2 expression was highest in the stratum pyramidal (SP) and stratum oriens (SO) layers for PV-ChR2 mice with cell bodies principally located in SP (Fig. 1a). Conversely, ChR2 expression was highest in the SO and stratum lacunosum moleculare (SLM) layers for SST-ChR2 mice with cell bodies principally located in SO (Fig. 1b). These expression profiles are consistent with the established roles of PV and SST interneurons providing perisomatic and dendritic inhibition, respectively[1,2,43]. To further confirm the spatially distinct inhibitory targets, we recorded interneuron subtype-specific inhibitory currents onto CA1 pyramidal neurons by activating ChR2 with 470 nm blue light (Fig. 1c). The rise and decay kinetics of the resulting light-evoked PV IPSCs were significantly faster compared to SST-derived IPSCs (Fig. 1d, e) (rise time: $3.7 \pm 0.3$ ms PV versus $6.9 \pm 0.7$ ms SST, $n = 14$, $p < 0.001$; decay time: $16 \pm 0.9$ ms PV versus $27 \pm 2.3$ ms SST, $n = 14$, $p < 0.001$) supporting a more proximal location for PV synapses compared to SST synapses. Light-evoked PV-IPSC kinetics were almost identical to IPSC kinetics recorded from paired whole-cell recordings made from PV expressing fast-spiking basket cells to CA1 pyramidal cells (Supplementary Fig. 1a) and similarly, selective activation of oriens lacunosum moleculare (OLM) cells using Chrna2-cre mice[44,45] revealed IPSC kinetics indistinguishable from SST-ChR mice (Supplementary Fig. 1e). Furthermore, measurement of synaptic response amplitudes as light was targeted to different regions of the slice confirmed the immunohistochemical characterisation of ChR2 expression supporting the selective stimulation of perisomatic versus dendritic targeted inhibition (Supplementary Fig. 1b, c, f, g). Therefore, whilst we cannot exclude the activation of other interneuron subtypes expressing PV or SST, these data suggest the majority of our synaptic inputs most likely arise from activation of PV basket cells and SST OLM cells that selectively target synapses to perisomatic and distal dendritic regions of CA1 pyramidal cells, respectively.

Having established two populations of inhibitory synapses, we investigated whether PV or SST synapses undergo long-term inhibitory synaptic plasticity and if so, whether induction and expression is similar at each synapse. IPSCs were recorded from CA1 pyramidal neurons held at 0 mv with glutamatergic transmission pharmacologically blocked. Interneuron subtype-specific IPSCs mediated by $GABA_A$ receptors were evoked by 5 ms light pulses and, importantly, an independent IPSC control pathway was evoked by electrical stimulation in the pyramidal layer (PV IPSCs) or stratum radiatum (SST IPSCs) (Fig. 1f and Supplementary Fig. 1d, h). Both PV and SST interneurons are entrained to theta frequency rhythms in the hippocampus[3] so we first tested whether bursts of IPSCs delivered at theta frequency could induce long-term inhibitory synaptic plasticity. Theta burst

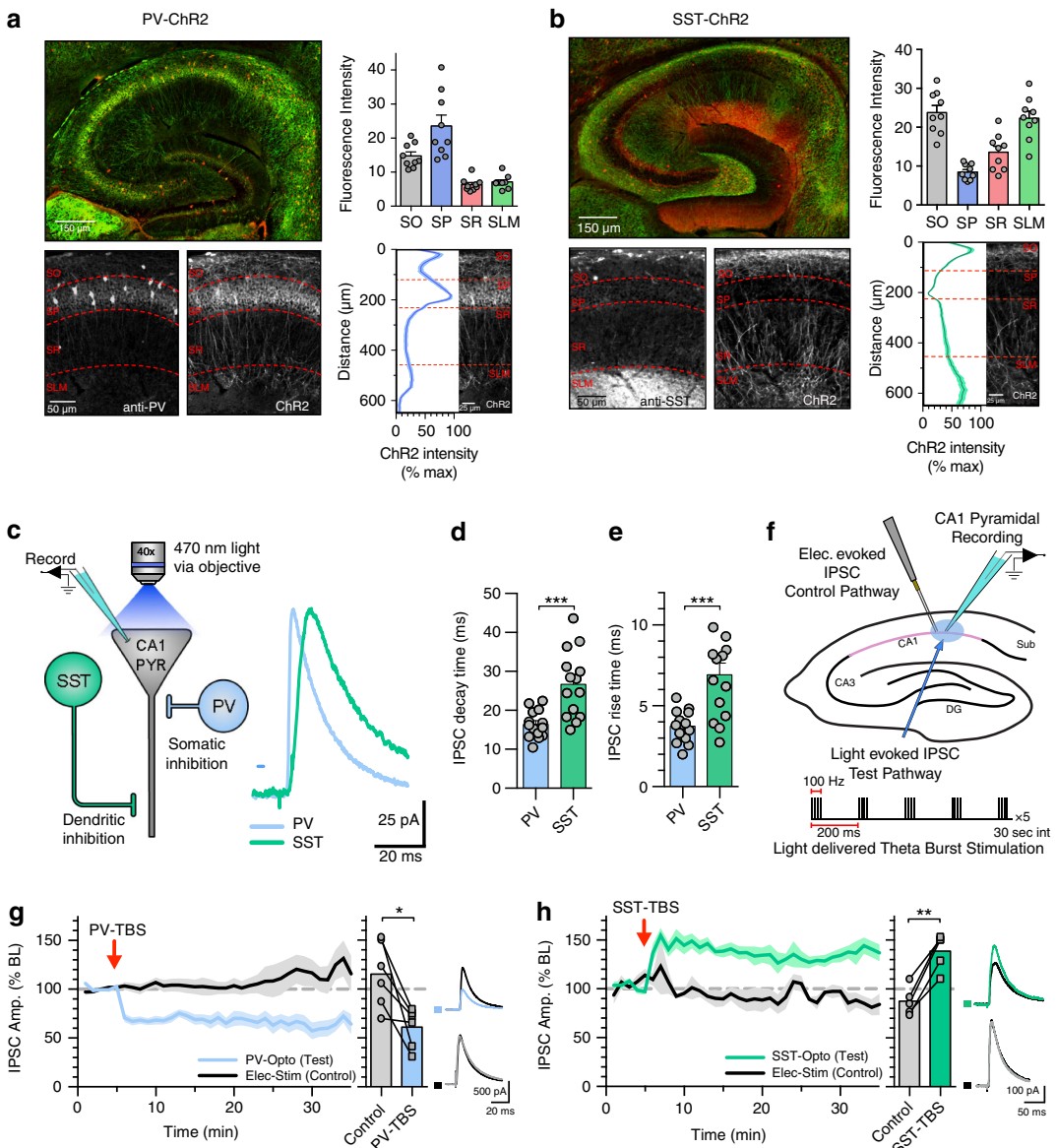

**Fig. 1 Somatically targeting PV and dendritically targeting SST inhibitory synapses undergo long-term synaptic plasticity. a** Immunohistochemistry showing expression of PV (red) and ChR2 (green) in PV-ChR2 mice. Histogram displaying mean ChR2 fluorescence expression levels in different hippocampal layers: stratum oriens (SO), stratum pyramidal (SP), stratum radiatum (SR) and stratum lacunosum moleculare (SLM) (right top), ChR2 expression as a function of distance across hippocampal layers (right bottom) ($n = 9$ slices from three animals). **b** Same as (**a**) but for SST-ChR2 mice. **c** Schematic and example IPSC traces highlighting the somatic and distal targeting of PV and SST synapses. **d** Summary of IPSC decay times for PV and SST IPSCs ($p = 0.0003$, unpaired $t$-test, two tailed, $n = 14$ cells). **e** Summary of IPSC rise times for PV and SST IPSCs ($p = 0.0004$, unpaired $t$-test, two tailed, $n = 14$ cells). **f** Schematic displaying the recording set up for inhibitory plasticity experiments and the light-induced theta burst (TBS) induction protocol. **g** TBS induced iLTD at PV synapses (left) average plasticity at control and test pathways (middle) and example traces before and after plasticity (right) ($p = 0.015$, paired $t$-test, two tailed, $n = 6$ cells). **h** TBS induced iLTP at SST synapses (left) average plasticity at control and test pathways (middle) and example traces before and after plasticity (right) ($p = 0.0011$, paired $t$-test, two tailed, $n = 5$ cells). Data presented as mean values ± SEM. See also Supplementary Figs. 1 and 2.

stimulation (TBS) of the light-evoked pathway led to a prolonged pathway-specific reduction of PV-IPSC amplitude indicating a synapse-specific long-term depression of PV synapses (PV-iLTD, $115 \pm 14\%$ control versus $61 \pm 8\%$ test pathway, $n = 6$, $p < 0.05$) (Fig. 1g). In contrast, an identical light-induced TBS led to a pronounced pathway-specific long-term potentiation of SST-IPSC amplitude (SST-iLTP, $87 \pm 6\%$ control versus $139 \pm 8\%$ test pathway, $n = 6$, $p < 0.01$) (Fig. 1h). These findings indicate that high frequency inhibitory synaptic stimulation can induce

inhibitory synaptic plasticity at PV and SST synapses, but the direction of plasticity is diametrically opposite for the two different synapses.

A common feature of plasticity at inhibitory synapses is the requirement for postsynaptic depolarisation in conjunction with synaptic stimulation, despite the synaptic stimulation itself causing hyperpolarisation[15,32–39,41,42]. To test the requirement for depolarisation in PV-iLTD and SST-iLTP, we delivered TBS whilst voltage clamping neurons at $-60$ mV during the TBS

protocol. Under these conditions neither PV-iLTD nor SST-iLTP were induced (Supplementary Fig. 2a, b; PV: $84 \pm 8\%$ control versus $91 \pm 4\%$ test pathway, $n = 5$, $p > 0.05$; SST: $96 \pm 6\%$ control versus $106 \pm 4\%$ test pathway, $n = 5$, $p > 0.05$), indicating that both forms of inhibitory plasticity require coincident pyramidal neuron depolarisation and inhibitory input.

**PV and SST synapses undergo spike timing-dependent plasticity.** During exploratory behaviour, neurons in the hippocampus are entrained to the theta rhythm[46] with defined populations of neurons, including different subpopulations of interneurons, firing action potentials at specific phases of the theta cycle[2–4]. Having established a requirement for coincident synaptic activity and postsynaptic depolarisation for the induction of PV-iLTD and SST-iLTP, we next sought to determine whether this was a Hebbian form of plasticity that could be induced by coincident pre- and postsynaptic action potentials and if so, what the precise spike timing requirements might be with respect to the preferred interneuron and pyramidal spiking phases of the theta cycle. The major subclasses of PV interneurons, basket cells and axo-axonic cells, fire on the descending phase of theta cycle roughly 60 ms before pyramidal neurons, whilst both bistratified and OLM SST interneurons fire near coincident with pyramidal neurons at the trough of the cycle (Fig. 2a)[2–4,6]. We therefore tested the induction of inhibitory spike timing-dependent plasticity (iSTDP) using spike timings of $-60$, $0$ and $+60$ ms to replicate spike patterns during exploratory behaviour and span the full width of a theta cycle (Fig. 2a).

To test iSTDP, CA1 pyramidal neurons were voltage clamped at $-50$ mV in the presence of AMPA and NMDA receptor blockers and light and electrically evoked IPSCs recorded as test and control synaptic pathways, respectively. During the iSTDP protocol, recordings were switched to current clamp with the membrane potential maintained at $-50$ mV and single light-evoked IPSCs were paired with a burst of four action potentials repeated 100 times at theta frequency (5 Hz). Stimulation of PV synapses with the theta relevant $-60$ ms spike timing protocol resulted in pathway-specific PV-iLTD (Fig. 2b, $95 \pm 7\%$ control versus $73 \pm 7\%$ test pathway, $n = 7$, $p < 0.05$). PV-iLTD was also observed when PV synapses were paired at 0 ms (Fig. 2c, $106 \pm 3\%$ control versus $72 \pm 6\%$ test pathway, $n = 13$, $p < 0.001$) but not at $+60$ ms (Fig. 2d, $105 \pm 6\%$ control versus $95 \pm 8\%$ test pathway, $n = 8$, $p > 0.05$) resulting in a pan-theta cycle iSTDP relationship incorporating iLTD but no iLTP (Fig. 2e). Surprisingly, SST synapses also displayed iLTD at $-60$ ms spike timings (Fig. 2f, $101 \pm 7\%$ control versus $70 \pm 5\%$ test pathway, $n = 6$, $p < 0.01$), but contrastingly underwent iLTP when paired at 0 ms (Fig. 2g, $104 \pm 4\%$ control versus $130 \pm 13\%$ test pathway, $n = 11$, $p < 0.05$). At pairings of $+60$ ms SST synapses also exhibited no plasticity similar to PV synapses (Fig. 2h, $92 \pm 8\%$ control versus $90 \pm 8\%$ test pathway, $n = 7$, $p > 0.05$). Therefore, SST synapses can undergo both iLTD and iLTP depending on the precise spike timing of pre- and postsynaptic action potentials in contrast to PV synapses that only undergo iLTD. These results demonstrate that spike timings observed during theta rhythm entrainment lead to distinct rules for iSTDP at PV and SST synapses. Interestingly, we observed at both PV and SST synapses that pairing inhibitory inputs 60 ms after a burst of action potentials was insufficient to induce inhibitory plasticity, highlighting the importance of spike timing and the need for inhibitory synaptic input prior to pyramidal neuron activity.

**Molecular mechanisms for PV-iLTD.** We next investigated the molecular mechanisms of spike timing-dependent PV-iLTD. First, we found that presynaptic input or postsynaptic spikes

alone were insufficient to induce iLTD at PV synapses (presynaptic input only: $100 \pm 12\%$ control versus $94 \pm 6\%$ test pathway, $n = 6$, $p > 0.05$; postsynaptic spikes only: $103 \pm 10\%$ control versus $94 \pm 5\%$ test pathway, $n = 6$, $p > 0.05$) (Fig. 3a, b). Consistent with PV-iSTDP and TBS induced PV-iLTD, these results show that coincident activity between PV interneurons and pyramidal neurons is required for PV-iLTD. Many forms of synaptic plasticity also depend on elevations in postsynaptic $Ca^{2+}$ and we tested if this was the case for PV-iLTD by including the $Ca^{2+}$ chelator BAPTA in the intracellular recording solution. BAPTA prevented PV-iLTD demonstrating a dependence on postsynaptic $Ca^{2+}$ ($100 \pm 10\%$ control versus $95 \pm 8\%$ test pathway, $n = 6$, $p > 0.05$) (Fig. 3c).

Important sources of postsynaptic $Ca^{2+}$ for the induction of excitatory and inhibitory synaptic plasticity are NMDA receptors and voltage-gated $Ca^{2+}$ channels (VGCCs)[41,47–49]. Since NMDA receptors are blocked in our experiments, we investigated the role of VGCCs in PV-iLTD. L-type VGCCs are the most prominent postsynaptic VGCCs, however, the L-type VGCC inhibitor nimodipine (20 µM) failed to block PV-iLTD ($97 \pm 4\%$ control versus $82 \pm 3\%$ test pathway, $n = 10$, $p < 0.01$; Fig. 3d). We next tested the role of T-type VGCCs using the inhibitor mibefradil (5 µM) which blocked PV-iLTD ($92 \pm 5\%$ control versus $92 \pm 3\%$ test pathway, $n = 6$, $p > 0.05$; Fig. 3e) and this was confirmed with the use of another T-type VGCC inhibitor ML218 (3 µM)[50] ($99.4 \pm 5.3\%$ control versus $93.8 \pm 7.2\%$ test pathway, $n = 8$, $p > 0.05$) (Supplementary Fig. 3e). Interestingly, T-type VGCCs have a low voltage threshold for activation and predominantly reside in an inactivated state at resting membrane potentials[51]. They therefore require hyperpolarisation to relieve voltage inactivation (de-inactivation), which corresponds precisely with the requirement for inhibitory synaptic input prior to postsynaptic depolarisation resulting in synapse specificity of PV-iLTD. These findings highlight a mechanism by which inhibitory synapses can provide a synapse-specific source of $Ca^{2+}$ to induce inhibitory plasticity.

The downstream effects of $Ca^{2+}$ can lead to release of retrograde signalling molecules, which regulate presynaptic release of GABA[36,40] or it can signal postsynaptically to reduce $GABA_A$ receptor function[32]. We therefore tested whether previously described retrograde signalling molecules nitrous oxide[40] and endocannabinoids[36] are involved in PV-iLTD. The nitrous oxide pathway antagonist ODQ (5 µM) and the CB1 receptor antagonist AM251 (1 µM) both failed to prevent PV-iLTD (Supplementary Fig. 3c, d and Fig. 3g, ODQ: $100 \pm 8\%$ control versus $78 \pm 7\%$ test pathway, $n = 6$, $p < 0.05$; AM251: $112 \pm 9\%$ control versus $77 \pm 7\%$ test pathway, $n = 6$, $p < 0.05$). An additional candidate mechanism could be the activation of the G-protein coupled $GABA_B$ receptor since this has been shown to mediate a form of iLTD[38], however, the $GABA_B$ receptor antagonist CGP 55845 (1 µM) also failed to prevent PV-iLTD (Supplementary Fig. 3b and Fig. 3g, $101 \pm 10\%$ control versus $71 \pm 7\%$ test pathway, $n = 7$, $p < 0.05$). One postsynaptic signalling pathway that has been implicated in iLTD is activation of the phosphatase calcineurin[32,52,53]. Application of the calcineurin inhibitor FK506 (10 µM) prevented PV-iLTD ($110 \pm 7\%$ control versus $105 \pm 3\%$ test pathway, $n = 7$, $p > 0.05$) (Fig. 3f), indicating a postsynaptic target for $Ca^{2+}$ signalling. In summary, PV-iLTD requires coincident pre- and postsynaptic activity, opening T-type VGCC to provide a postsynaptic $Ca^{2+}$ signal that leads to activation of calcineurin to induce LTD at PV inhibitory synapses (Figs. 3g and 5a).

**Molecular mechanisms for SST-iLTP.** The molecular mechanisms of spike timing-dependent SST-iLTP were next investigated. Similar to PV-iLTD, and consistent with SST-iSTDP and TBS

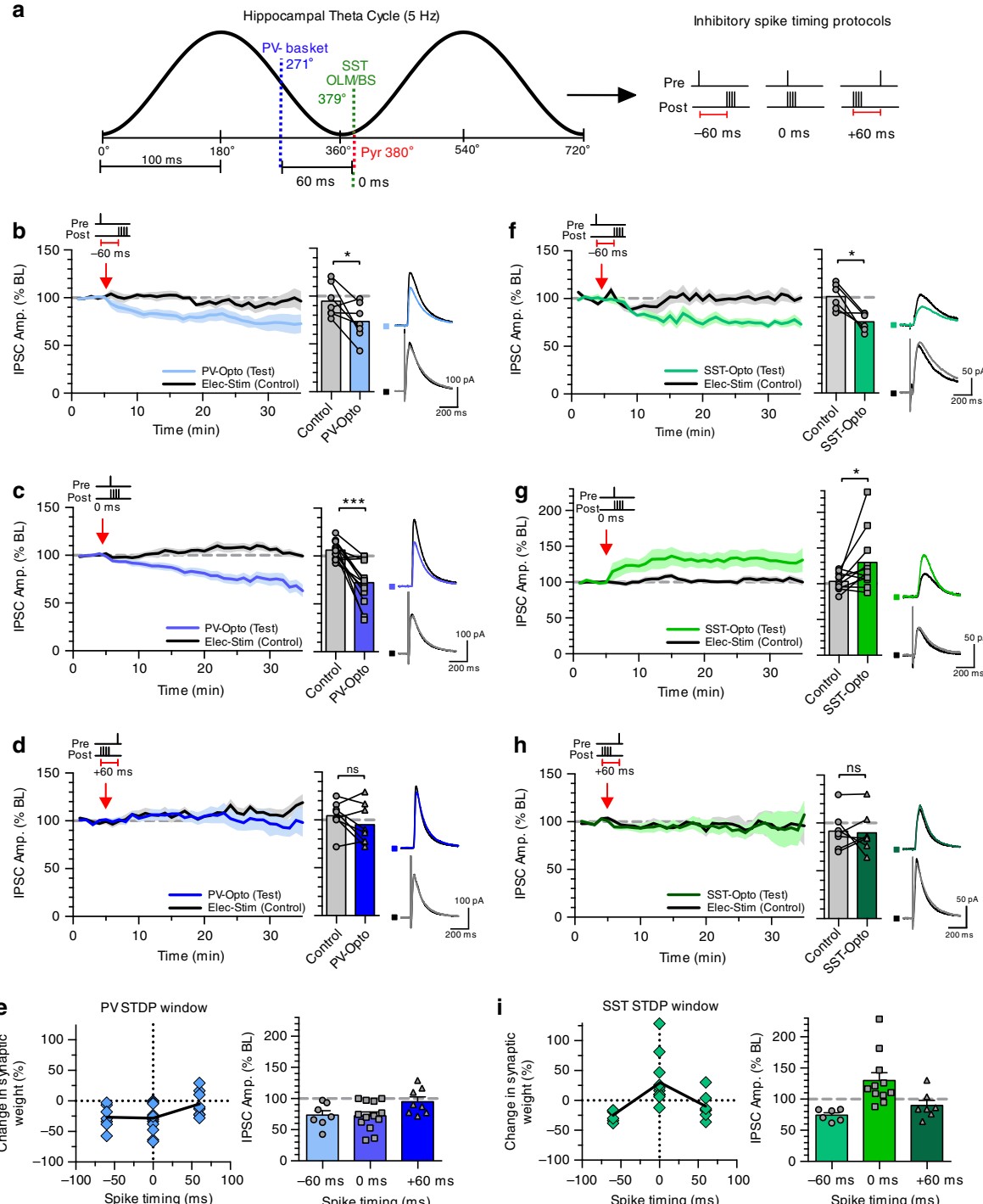

**Fig. 2 PV and SST inhibitory synapses undergo spike timing-dependent plasticity. a** Schematic highlighting the relative spike timing of PV and SST expressing interneurons during theta oscillations in relation to pyramidal neuron spiking (left). The three pairing protocols used for iSTDP experiments representing presynaptic stimulation and postsynaptic action potentials −60 ms pre before post, 0 ms pre and post together and +60 ms post before pre (right). **b** −60 ms pre before post pairing induced iLTD at PV synapses (left) average plasticity at control and test pathways (middle) and example traces for before and after plasticity (right) ($p = 0.0401$, paired $t$-test, two tailed, $n = 7$ cells). **c** 0 ms pre and post pairing induced iLTD at PV synapses (left) average plasticity at control and test pathways (middle) and example traces for before and after plasticity (right) ($p = 0.0003$, paired $t$-test, two tailed, $n = 13$ cells). **d** +60 ms post before pre pairing failed to induce plasticity at PV synapses (left) average plasticity at control and test pathways (middle) and example traces for before and after plasticity (right) ($p = 0.1682$, paired $t$-test, two tailed, $n = 8$ cells). **e** Summary of the inhibitory STDP window at PV synapses. **f** Same as (**b**) but for SST synapses ($p = 0.0099$, paired $t$-test, two tailed, $n = 6$ cells). **g** Same as (**c**) but for SST synapses ($p = 0.0481$, paired $t$-test, two tailed, $n = 11$ cells). **h** Same as (**d**) but for SST synapses ($p = 0.8315$, paired $t$-test, two tailed, $n = 7$ cells). **i** Same as (**e**) but for SST synapses. Data presented as mean values ± SEM. Scale bars (**b**–**d**): 200 ms, 100 pA; **f**–**h** 200 ms, 50 pA.

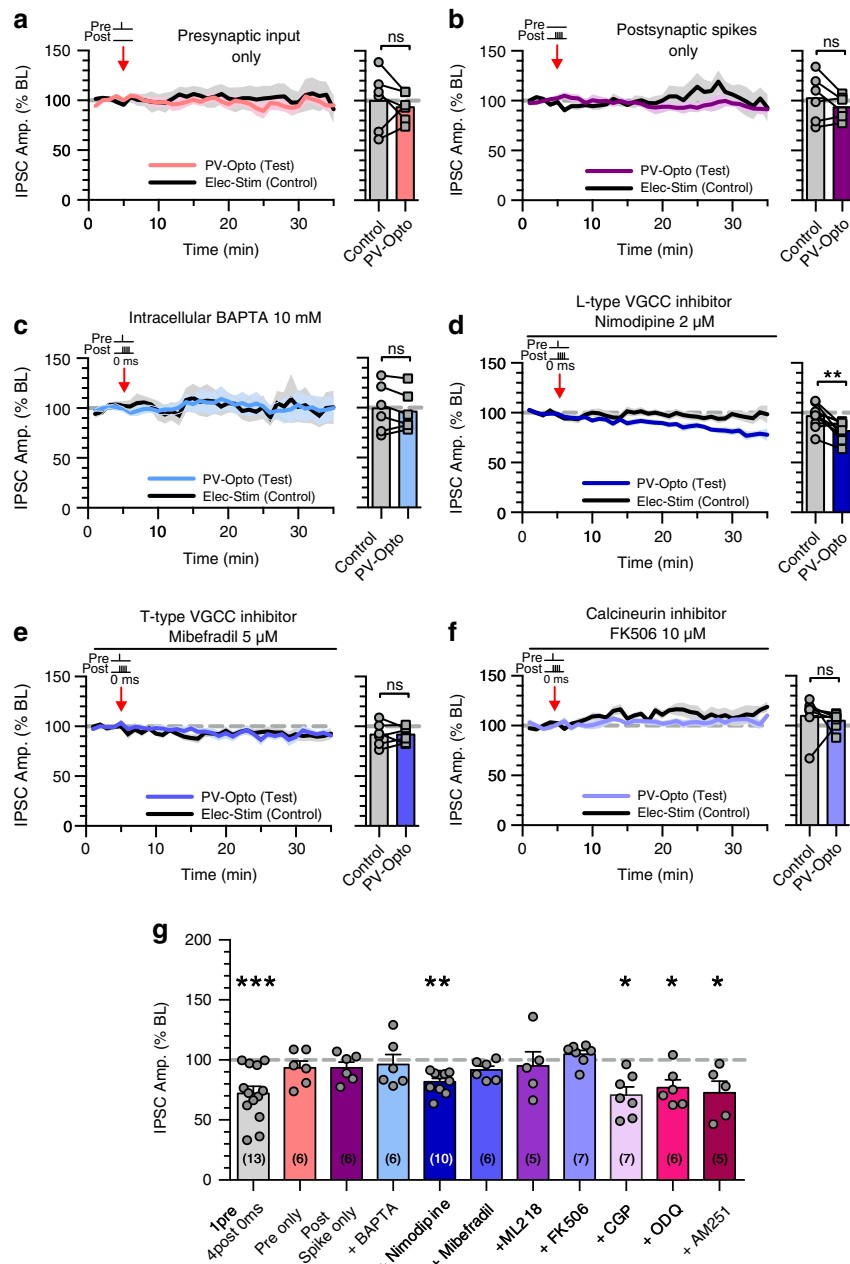

**Fig. 3 PV-iLTD requires activation of T-type VGCCs and calcineurin. a** Presynaptic stimulation of PV inputs alone induced no plasticity ($p = 0.530$, paired $t$-test, two tailed, $n = 6$ cells). **b** Postsynaptic spikes alone failed to induce plasticity at PV synapses ($p = 0.208$, paired $t$-test, two tailed, $n = 6$ cells). **c** Inclusion of BAPTA in the internal recording solution occludes PV-iLTD upon 0 ms pre and post pairing ($p = 0.366$, paired $t$-test, two tailed, $n = 6$ cells). **d** L-type calcium channel antagonist nimodipine does not block PV-iLTD upon 0 ms pre and post pairing ($p = 0.0036$, paired $t$-test, two tailed, $n = 10$ cells). **e** T-type calcium channel antagonist mibefradil occludes PV-iLTD upon 0 ms pre and post pairing ($p = 0.959$, paired $t$-test, two tailed, $n = 6$ cells). **f** Calcineurin inhibitor, FK506 occludes PV-iLTD upon 0 ms pre and post pairing ($p = 0.545$, paired $t$-test, two tailed, $n = 7$ cells). In panels (**a–f**), average plasticity in control and test pathways is shown on the right. **g** Summary histogram displaying the level of plasticity under each experimental condition significance refers to paired $t$-tests in (**a–f**). Data presented as mean values ± SEM. See also Supplementary Fig. 3.

induced SST-iLTP, we found that SST-iLTP requires coincident pre- and postsynaptic activation as neither SST inputs nor postsynaptic action potentials alone were able to induce SST-iLTP (presynaptic input only: $107 \pm 6\%$ control versus $91 \pm 11\%$ test pathway, $n = 6$, $p > 0.05$; postsynaptic spikes only: $94 \pm 5\%$ control versus $100 \pm 10\%$ test pathway, $n = 7$, $p > 0.05$) (Fig. 4a, b). The inclusion of the $Ca^{2+}$ chelator BAPTA also prevented the induction of iLTP at SST synapses ($97 \pm 10\%$ control versus $92 \pm 6\%$ test pathway, $n = 6$, $p > 0.05$) (Fig. 4c), indicating SST-iLTP requires postsynaptic $Ca^{2+}$. Again, we assessed if L- and/or

T-type VGCCs could provide the source of postsynaptic $Ca^{2+}$ required for SST-iLTP. Interestingly, L-Type VGCC antagonist nimodipine and T-type VGCC antagonists mibefradil and ML218 both blocked SST-iLTP (nimodipine: $93 \pm 7\%$ control versus $101 \pm 7\%$ test pathway, $n = 5$, $p > 0.05$; mibefradil: $99 \pm 9\%$ control versus $101 \pm 10\%$ test pathway, $n = 7$, $p > 0.05$; ML218: $113 \pm 8\%$ control versus $100 \pm 6\%$ test pathway, $n = 7$, $p > 0.05$) (Fig. 4d, e, h and Supplementary Fig. 4b), showing SST-iLTP requires activation of L-type and T-type VGCCs with the latter providing a synapse-specific source of $Ca^{2+}$ similar to PV-iLTD. In addition,

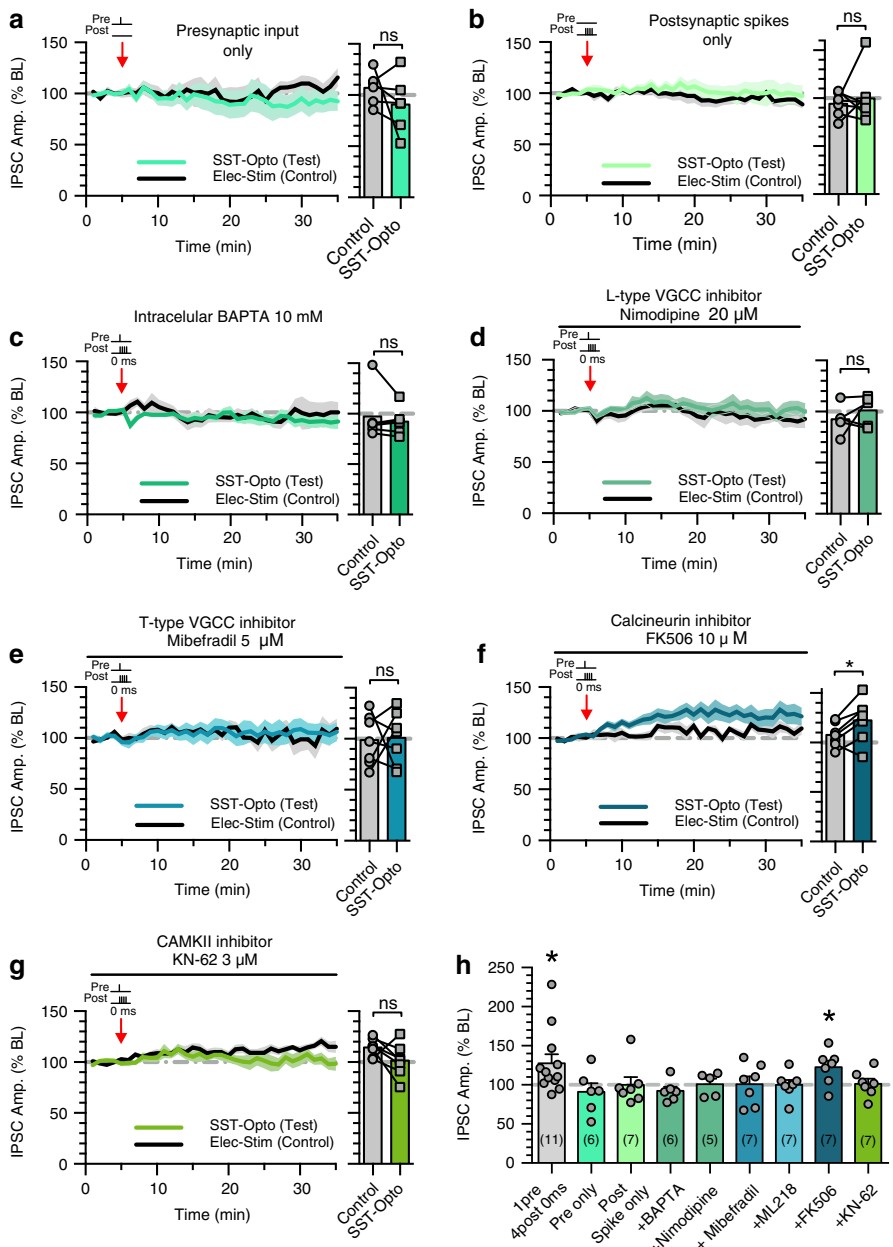

**Fig. 4 SST-iLTP requires activation of L-type and T-type VGCCs and CAMKII. a** Presynaptic stimulation of SST inputs alone failed to induce plasticity ($p = 0.279$, paired $t$-test, two tailed, $n = 6$ cells). **b** Postsynaptic spikes alone failed to induce plasticity at SST synapses ($p = 0.623$, paired $t$-test, two tailed, $n = 7$ cells). **c** Inclusion of BAPTA in the internal recording solution occludes SST-iLTP upon 0 ms pre and post pairing ($p = 0.385$, paired $t$-test, two tailed, $n = 6$ cells). **d** L-type calcium channel antagonist nimodipine occludes SST-iLTP upon 0 ms pre and post pairing ($p = 0.379$, paired $t$-test, two tailed, $n = 5$ cells). **e** T-type calcium channel antagonist mibefradil occludes SST-iLTP upon 0 ms pre and post pairing ($p = 0.899$, paired $t$-test, two tailed, $n = 7$ cells). **f** Calcineurin inhibitor, FK506 fails to block SST-iLTP upon 0 ms pre and post pairing ($p = 0.028$, paired $t$-test, two tailed, $n = 7$ cells). **g** CAMKII inhibitor KN-62 occludes SST-iLTP upon 0 ms pre and post pairing ($p = 0.120$, paired $t$-test, two tailed, $n = 7$ cells). In panels (**a–g**), average plasticity in control and test pathways is shown on the right. **h** Summary histogram displaying the level of plasticity under each experimental condition significance refers to paired $t$-tests in (**a–g**). Data presented as mean values ± SEM. See also Supplementary Fig. 4.

preventing T-type VGCC de-inactivation via shifting the inhibitory current polarity from hyperpolarising to depolarising (via a higher internal chloride concentration), also prevented SST-iLTP ($83 \pm 15\%$ control versus $90 \pm 5\%$ test pathway, $n = 6$, $p > 0.05$) (Supplementary Fig. 4c) further highlighting the role of T-type VGCC. Finally the T-type VGCC antagonist ML218 also blocked TBS induced SST-iLTP ($93 \pm 7\%$ control versus $89 \pm 7\%$ test pathway, $n = 5$, $p > 0.05$) (Supplementary Fig. 4d), indicating that

these plasticity mechanisms share a similar molecular mechanism.

Since calcineurin is required for PV-iLTD, we next tested the possible involvement of calcineurin in SST-iLTP. However, inhibition of calcineurin failed to block SST-iLTP ($108 \pm 5\%$ control versus $123 \pm 8\%$ test pathway, $n = 7$, $p < 0.05$) (Fig. 4f, h). As SST-iLTP requires both L-type and T-type VGCCs, we hypothesised that SST-iLTP might be mediated by a molecular pathway engaged by high levels of postsynaptic $Ca^{2+}$. CAMKII is

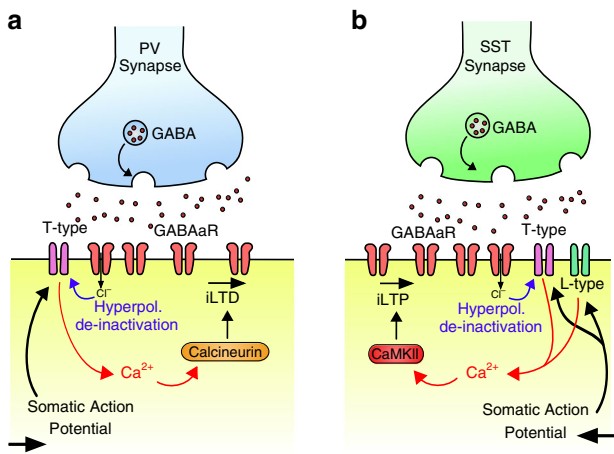

**Fig. 5 PV-iLTD and SST-iLTP mechanisms. a** Illustration of PV-iLTD mechanism. Hyperpolarisation by GABA$_A$ receptor currents relieves T-type VGCCs from voltage dependent inactivation (de-inactivation). Back-propagating action potentials then activate T-type VGCCs providing an inhibitory synapse-specific source of Ca$^{2+}$ to activate calcineurin resulting in LTD at PV synapses. **b** Illustration of SST-iLTP mechanism. Similar to PV-iLTD but requiring activation of T-type and L-type VGCCs that activates CAMKII resulting in LTP at SST synapses.

one such candidate and has been shown to mediate potentiation of inhibitory synapses including SST synapses within the cortex[41] and other inhibitory synapses within the hippocampus[54,55] resulting in postsynaptic changes in GABA$_A$ receptors. We therefore tested the involvement of CAMKII activation on hippocampal SST-iLTP and found that the CAMKII inhibitor KN-62 (3 μM) prevented SST-iLTP (115 ± 3% control versus 102 ± 6% test pathway, $n = 7$, $p > 0.05$) (Fig. 4g, h), consistent with its role in mediating iLTP. In summary, SST-iLTP requires the coincident activation of SST synapses and pyramidal neurons, which activates L-type and T-type VGCCs providing a Ca$^{2+}$ source able to activate CAMKII to induce SST-iLTP (Fig. 5b).

**SST and PV plasticity shapes pyramidal neuron output.** To understand the potential implications of PV-iLTD and SST-iLTP on network integration of inputs to a pyramidal neuron, we implemented a multi-compartment model of a CA1 pyramidal neuron in the presence of proximal (PV) and distal (SST) inhibition (Fig. 6a). The simulated CA1 pyramidal neuron receives distal excitatory input from entorhinal inputs via the temporoammonic (TA) pathway and proximal excitatory input from CA3 inputs via the Schaffer collateral (SC) pathway. We also implemented rate-based inhibitory plasticity rules derived from our experiments under the most physiologically relevant conditions (SST-iLTP and PV-iLTD). Inhibitory synaptic weights onto pyramidal cells were therefore updated following a Hebbian plasticity rule in which coincident pre- and postsynaptic activity leads to iLTD for PV synapses and iLTP for SST synapses. We then simulated activity within the network before and after the induction of inhibitory plasticity and compared the correlation between SC or TA inputs and pyramidal cell activity at these two stages. As expected, the correlation between SC inputs and CA1 pyramidal cell activity increased and the correlation between TA inputs and CA1 pyramidal cell activity decreased following the induction of interneuron plasticity (Fig. 6b). Therefore, if we assume that SST-iLTP occurs primarily at distal dendritic locations, interneuron-specific plasticity is a potential mechanism to change CA1 network state from being driven by both TA and SC inputs to being primarily driven by SC inputs.

We next incorporated functionally relevant feedforward and feedback connectivity for PV and SST interneurons within the CA1 network. PV interneurons receive strong feedforward innervation from the SC pathway but relatively limited input from the TA pathway and some feedback input from CA1 pyramidal neurons[7,9,43,56]. In contrast, distally targeting SST interneurons receive almost no feedforward input and are driven by feedback input from CA1 pyramidal neurons[9,43,57,58]. There is also evidence that bistratified interneurons that target inhibition to proximal dendrites in stratum radiatum and express both PV and SST can be feedforward in the SC pathway and also feedback within CA1[9,43].

Using these various functional connectivity arrangements, we first investigated the consequences of PV-iLTD on CA1 pyramidal cell output. For SC inputs with feedforward PV interneurons, PV-iLTD led to an increase in CA1 pyramidal cell activity due to a reduction in feedforward inhibition (Fig. 6c). The same result was achieved if we included feedback inhibition from PV interneurons (Supplementary Fig. 5a). When we considered the TA pathway without feedforward PV interneurons, PV-iLTD did not change pyramidal cell output (Fig. 6d). However, if we incorporated PV interneurons as feedforward and feedback inhibition, PV-iLTD led to an increase in pyramidal cell activity in response to TA stimulation (Supplementary Fig. 5b). Therefore, our model predicts that CA1 pyramidal cell activity in response to SC stimulation should increase following PV-iLTD, whereas it is likely to remain unchanged in response to stimulation of the TA pathway unless PV interneurons participating in feedforward inhibition of the TA pathway or feedback inhibition are significantly activated and undergo PV-iLTD.

We next investigated the implications of SST-iLTP on CA1 pyramidal cell output. Pyramidal cell activity in response to SC stimulation was not affected by SST-iLTP if SST synapses were located at the pyramidal cell's distal dendritic compartment (Fig. 6e) and this was true regardless of whether SST interneurons were activated in feedforward or feedback fashion (Supplementary Fig. 5c). Contrary to SC stimulation, TA-induced CA1 pyramidal cell activity was reduced after SST-iLTP (Fig. 6f) and this effect was stronger if SST interneurons were considered to be feedforward as well as feedback (Supplementary Fig. 5d). In summary, our model predicts that SST-iLTP does not affect SC-induced CA1 pyramidal cell activity, whereas SST-iLTP decreases activity induced by TA stimulation.

If we assume the functional connectivity shown in Fig. 6c–f, our model simulations therefore predict that PV-iLTD will increase CA1 pyramidal neuron responses to SC but not TA inputs and that SST-iLTP will decrease CA1 pyramidal neuron responses to TA but not SC inputs.

To test these predictions, we experimentally investigated the impact of PV-iLTD and SST-iLTP on the probability of spike generation in CA1 pyramidal neurons. By stimulating either the SC or TA pathways, action potential probability was recorded in response to 10 consecutive EPSPs where the stimulation intensity was set such that the baseline action potential probability for each EPSP was ~ 50%. Upon SC stimulation, PV-iLTD (0 ms timing) led to an increase in the spike probability (0.4 ± 0.07 baseline versus 0.78 ± 0.05 post plas, $n = 6$, $p < 0.05$) (Fig. 7a) that mirrored the timecourse of PV-iLTD, but for TA pathway stimulation, spike probability was unaltered (0.44 ± 0.04 baseline versus 0.47 ± 0.03 post plas, $n = 6$, $p > 0.05$) (Fig. 7b). Taken together, these results confirm the predictions from the model and suggest that in our experiments the majority of PV interneurons recruited by SC stimulation are feedforward and undergo PV-iLTD, whereas few PV interneurons are recruited by TA stimulation or via feedback excitation from CA1 pyramidal

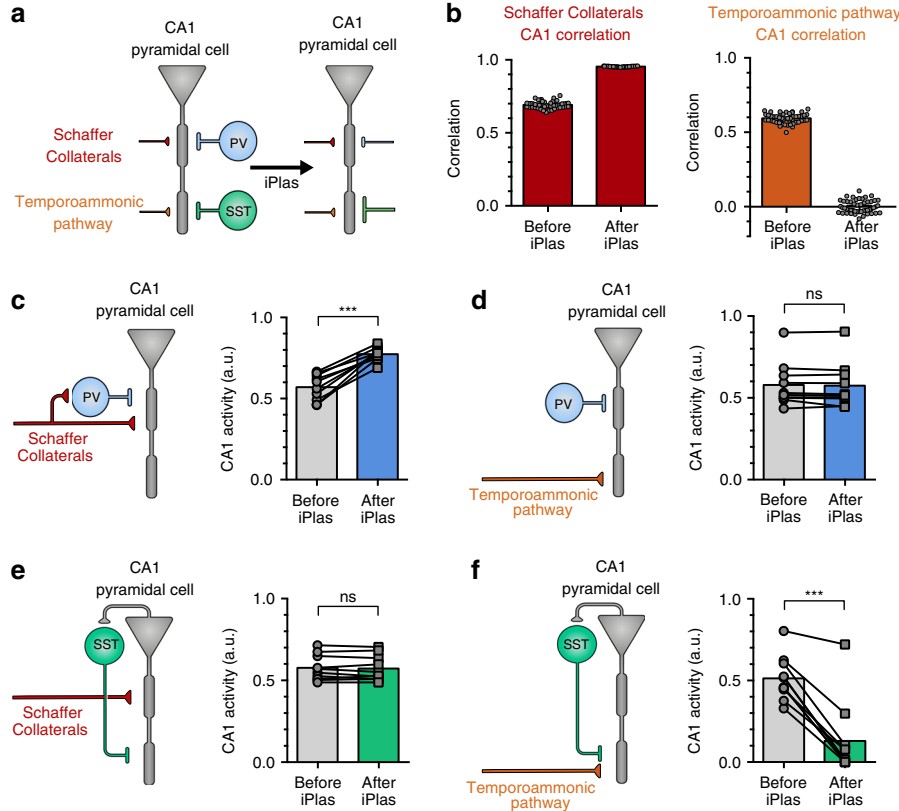

**Fig. 6 PV and SST plasticity differentially regulates Schaffer collateral and temporoammonic excitation of CA1 pyramidal neurons. a** Diagram of a simulated, rate-based CA1 pyramidal cell before and after the induction of inhibitory plasticity (iPlas). A single two-compartment neuron receives inputs from four sources: distally targeting temporoammonic, proximally targeting Schaffer collaterals, distally targeting inhibition from SST interneurons, and proximally targeting inhibition from PV interneurons. iPlas leads to PV-iLTD and SST-iLTP. **b** Correlation between Schaffer collateral activity and CA1 somatic activity (left) and temporoammonic activity and CA1 somatic activity (right) before and after iPlas at PV and SST synapses (PV-iLTD and SST-iLTP). **c–f** CA1 somatic activity before and after iPlas (either PV-iLTD or SST-iLTP) under the individual stimulation of either Schaffer collaterals or temporoammonic inputs. **c** Schaffer collateral-driven CA1 somatic activity is enhanced upon PV-iLTD ($p < 0.0001$, paired $t$-test, two tailed, $n = 10$). **d** CA1 somatic activity driven by temporoammonic input after PV-iLTD is unchanged ($p = 0.391$, paired $t$-test, two tailed, $n = 10$). **e** CA1 somatic activity driven via Schaffer collateral input after SST-iLTD is unchanged ($p = 0.572$, paired $t$-test, two tailed, $n = 10$). **f** SST-iLTP leads to a reduction in CA1 somatic activity in response to temporoammonic input ($p < 0.0001$, paired $t$-test, two tailed, $n = 10$). Data presented as mean values ± SEM. See also Supplementary Fig. 5.

neurons. We next conducted experiments to examine the impact of SST-iLTP on spike generation. We found that the increase in SST inhibition with SST-iLTP had little effect on action potential generation from SC stimulation ($0.53 \pm 0.04$ baseline versus $0.5 \pm 0.08$ post plas, $n = 8$, $p > 0.05$) (Fig. 7c) but led to a robust reduction in spike generation from the TA pathway ($0.52 \pm 0.03$ baseline versus $0.27 \pm 0.09$ post plas, $n = 6$, $p < 0.05$; Fig. 7d). Again, these results confirm the predictions from the model and suggest that in our experiments SST interneurons are primarily feedback and target distal dendritic regions of pyramidal neurons. Furthermore, analysis to separate differential changes in spike probability across the ten stimulus train showed that spike probability for TA responses later in the train were reduced after SST-iLTP, whereas those earlier in the train were not. This might be expected for an increase in feedback inhibition, which is engaged more during the later responses (Supplementary Fig. 6c). In contrast, the changes in spike probability for SC responses were even across all responses after PV-iLTD, consistent with an equal reduction in feedforward inhibition for all of the ten responses (Supplementary Fig. 6b).

Taken together, these results demonstrate that long-term inhibitory plasticity changes the responses of CA1 pyramidal neurons prioritising inputs from the SC pathway over those from the TA pathway. Increased distal dendritic inhibition driven by SST-iLTP will also inhibit the induction of excitatory synaptic plasticity[44,59] with important functional implications for the formation and stability of place cells.

**SST and PV plasticity enables place cell stability**. We next explored the implications of long-term inhibitory plasticity on place cell physiology within hippocampal networks. The long-term nature of inhibitory plasticity suggests that its impact on place cell physiology will be evident as an animal traverses different environments. Key features of place cells are: (1) that in multiple different environments each place cell may represent distinct locations or switch to be silent, and (2) that within any single environment, place cell representations are broadly stable upon repeated exposures to that environment[22,60]. However, these two features are somewhat contradictory since they require place cells to respond to different inputs without interference[23].

To investigate the functional implications of interneuron subtype-dependent long-term plasticity for place cell physiology in multiple different environments, we simulated a CA1 network receiving place-tuned input, while an animal explored first an annular track (environment A) and then a different track (environment B) before finally returning to the original familiar environment A' (Fig. 8a). In our simulations, CA1 pyramidal

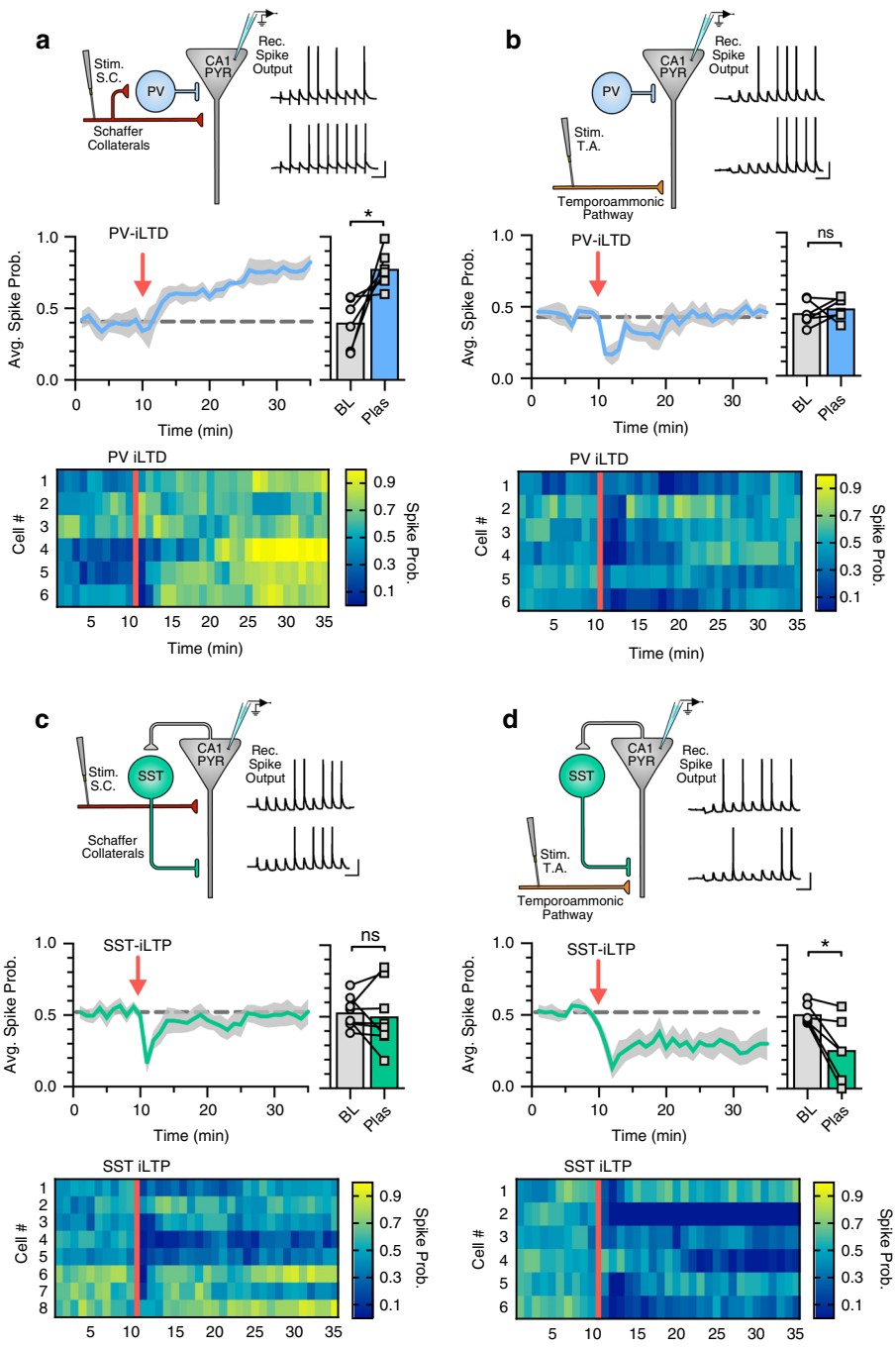

**Fig. 7 CA1 output driven by Schaffer collateral or temporoammonic inputs is differentially regulated by PV-iLTD and SST-iLTP. a** Diagram showing the experimental design where electrically stimulated Schaffer collaterals evoke action potentials in CA1 pyramidal neurons with example current-clamp traces before and after induction of PV-iLTD (0 ms pre and post pairing) (top). Spike probability timecourse (left) and average spike probability during baseline (BL) and 20–25 min after induction of PV-iLTD (Plas) (right) ($p = 0.0288$, paired $t$-test, two tailed, $n = 6$ cells). **b** Same as (**a**) but for the stimulation of the temporoammonic pathway ($p = 0.5264$, paired $t$-test, two tailed, $n = 6$ cells). **c** Diagram showing electrical stimulation of Schaffer collaterals with example traces before and after induction of SST-iLTP (0 ms pre and post pairing) (top). Spike probability timecourse (left) and average spike probability during baseline (BL) and after SST-iLTP (Plas) (right) ($p = 0.667$, paired $t$-test, two tailed, $n = 8$ cells). **d** Same as (**c**) but for stimulation of the temporoammonic pathway ($p = 0.0203$, paired $t$-test, two tailed, $n = 6$ cells). Data presented as mean values ± SEM. Scale bars: 100 ms, 20 mV. See also Supplementary Fig. 6.

neurons receive inputs from SCs, TA pathway, SST interneurons and PV interneurons (Fig. 8b). SC inputs are spatially tuned, while the other inputs are considered spatially uniform for simplicity. SC inputs are plastic and follow a Hebbian-type plasticity rule, which depends on coincident pre- and postsynaptic activation. Following recent evidence that place fields are

formed by synaptic plasticity at SC synapses following closely timed TA and SC inputs[19,61], we implemented the postsynaptic term of the Hebbian plasticity rule to be the product of the activities of the distal and proximal compartments of our two-compartment neuron model (see "Methods"). SST and PV inhibitory synapses onto pyramidal cells also follow a rate-

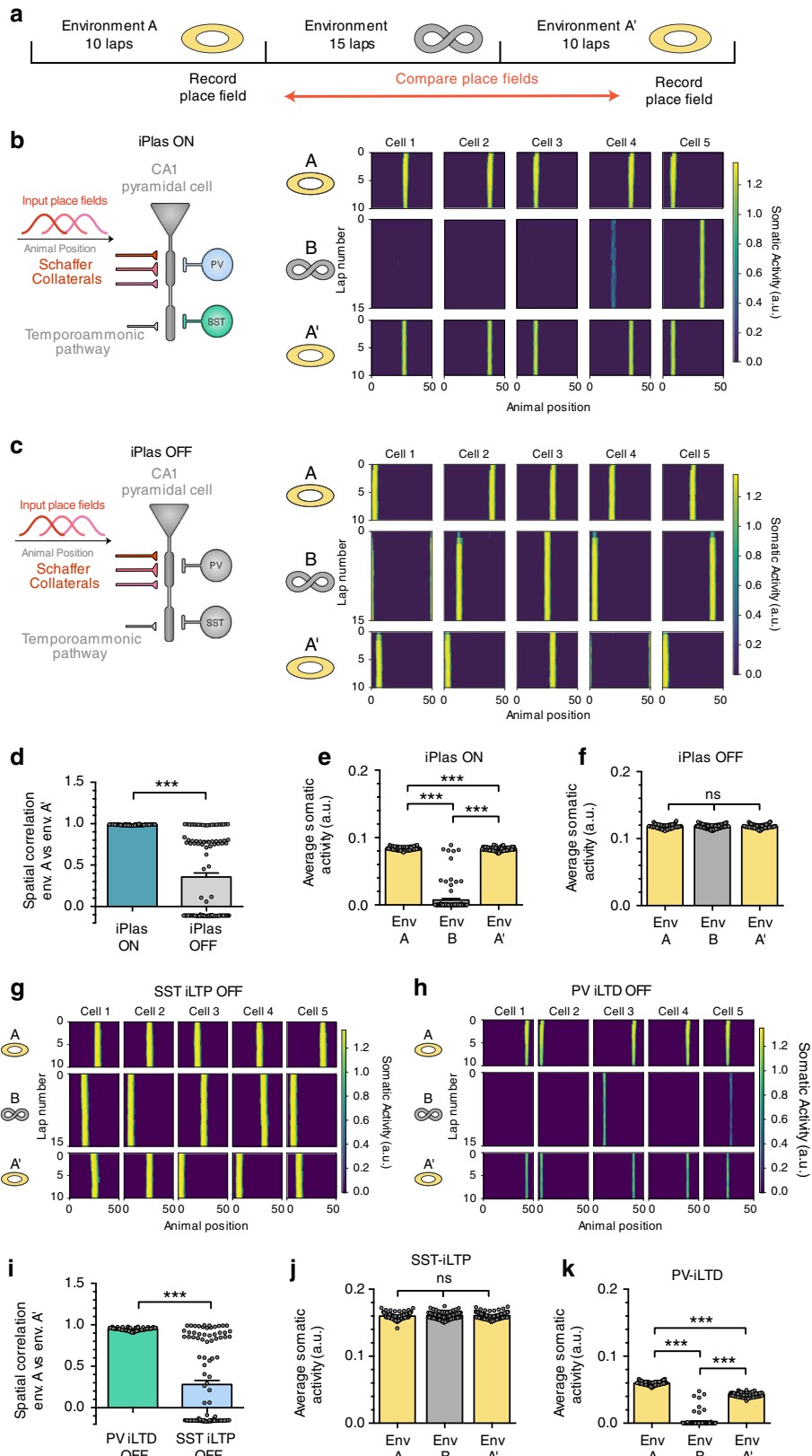

based Hebbian-type plasticity rule inspired by the physiologically relevant scenarios from our experimental data (as for Fig. 6). Coincident pre- and postsynaptic activity leads to iLTP in the case of SST synapses, whereas pre- and postsynaptic coactivation leads to iLTD in the case of PV synapses.

For any trial simulation of the network, the simulated CA1 pyramidal neuron rapidly developed a place field at a random location within environment A that remained stable for subsequent laps of the track (Fig. 8b). These place fields were driven by rapidly evolving synaptic weight changes (Supplementary Fig. 7). When the

**Fig. 8 PV and SST plasticity ensures place cell stability and fidelity across multiple environments. a** Simulation protocol. An animal explores environment A for ten laps. It is then moved to environment B and explores it for 15 laps. Finally, the animal is moved back to the first environment (A') and is allowed to run for another ten laps. Throughout this protocol, two-compartment CA1 pyramidal cells are simulated receiving spatially tuned Schaffer collateral inputs, temporoammonic input and PV and SST inhibitory inputs. Schaffer collateral synapses follow a Hebbian-type excitatory plasticity dependent on the coactivation of dendritic and somatic compartments (see "Methods"). PV and SST synapses undergo rate-based iPlas (PV-iLTD and SST-iLTP). We simulate the switch from environment A to environment B by randomly shuffling the identity of the SC inputs to the CA1 pyramidal neuron. Schematic depictions of environments A and B indicate their cyclical nature. **b** Diagram of simulated CA1 pyramidal cell and examples of somatic activity during exploration for iPlas ON. With iPlas (PV-iLTD and SST-iLTP) ON, place field location formed in environment A (top and bottom panel) remains stable after exposure to novel environment B (middle panel). **c** Diagram of simulated CA1 pyramidal cell and examples of somatic activity during exploration for iPlas OFF. Location of place fields formed in environment A is not maintained after exposure to a novel environment. **d** Spatial correlation between environment A before and after exposure to novel environment is maintained when iPlas is present but is reduced without iPlas ($p < 0.0001$, Mann–Whitney test, $n = 100$). **e** When iPlas is ON, average somatic activity of recently formed place cell is significantly reduced in new environment B but returns to higher levels when the animal returns to environment A' ($p < 0.0001$, Friedman test, $n = 100$). **f** When iPlas is OFF, average somatic activity remains high in both environments (A versus B, $p > 0.999$; A versus A', $p = 0.472$; B versus A', $p = 0.143$, Friedman test, $n = 100$). **g** When SST-iLTP is turned off, leaving just PV-iLTD, place cell locations are not retained after exposure to a new environment. **h** When PV-iLTD is turned off, leaving just SST-iLTP, place cell locations are maintained across multiple environments. **i** Spatial correlation between environment A before and after exposure to novel environment is maintained when only PV-iLTD is turned off but is reduced when only SST-iLTP is turned off ($p < 0.0001$, Mann–Whitney test, $n = 100$). **j** SST-iLTP is turned off, leaving just PV-iLTD, average somatic activity remains high in both environments (A versus B, $p = 0.967$; A versus A', $p = 0.774$; B versus A', $p > 0.999$, Friedman test, $n = 100$). **k** When PV-iLTD is turned off, leaving just SST-iLTP, average somatic activity is lower and thus less robust ($p < 0.0001$, Friedman test, $n = 100$). Data presented as mean values ± SEM. Statistical significance between groups was assessed via Dunn's multiple comparisons test. See also Supplementary Fig. 7.

track was switched to environment B, the indexes of Shaffer collateral inputs were shuffled. In this environment, the CA1 pyramidal neuron occasionally formed a new place field but was more often silent due to an inability to align and adapt synaptic weight increases after the inputs were shuffled (Fig. 8b and Supplementary Fig. 7a). On returning to the familiar environment A', the initial place field location was reinstated immediately (Fig. 8b, d, e). These results are qualitatively in line with the experimentally observed physiology of place cell activity in different environments[22].

To determine the role of inhibitory plasticity, we first removed PV-iLTD and SST-iLTP from the model. Simulations of this model lacking inhibitory plasticity showed similar place cell activity in environment A. In contrast, when the track was switched to environment B without inhibitory plasticity engaged, synaptic weight changes drove the generation of new place fields in every trial and the overall spiking rates were not reduced (Fig. 8c and Supplementary Fig. 7b). Furthermore, on returning to environment A, the place fields were no longer reinstated but instead new representations evolved (Fig. 8c, d, f). Thus, without inhibitory plasticity, novel environments generate interference and the network is no longer capable of creating stable place field representations.

We next sought to distinguish the roles of PV-iLTD and SST-iLTP within this network. Simulations of a model with only PV-iLTD (SST-iLTP OFF) showed similar lack of place cell stability across environments A–B–A' to simulations with no inhibitory plasticity and overall spiking rates were unchanged in environments B and A' due to prior spiking rate saturation (Fig. 8g, i, j). With implementation of only SST-iLTP (PV-iLTD OFF), place cell stability across environments A–B–A' was reinstated but overall spiking rates were reduced compared to simulations with full inhibitory plasticity (Fig. 8i, k).

These circuit-level modelling data show how long-term inhibitory plasticity can provide a mechanism for the experimentally observed phenomenon that newly formed place cells are stable with repeated exposure to an environment and don't undergo interference from experiencing other environments. This stability is principally due to SST-iLTP, which also reduces the efficiency of forming new place fields in different environments. The overall spike output of place cells is maintained by PV-iLTD which counteracts the reduction in spike output caused by SST-iLTP.

## Discussion

Inhibitory GABAergic synapses are known to undergo long-term plasticity, but very few studies have defined which subpopulations of inhibitory interneurons are engaged and whether plasticity is induced by physiological firing patterns. This is important since distinct interneuron subtypes play highly specific roles within neuronal networks[1] and therefore plasticity at one inhibitory synapse may have very different effects to another. In this study, we address this complexity and show that proximal and distal dendritically targeting interneuron synapses on CA1 pyramidal neurons have distinct plasticity rules within the hippocampus. These inhibitory synapses undergo homosynaptic plasticity in a Hebbian manner relying on the coincident activation of interneurons and pyramidal neurons (Figs. 1 and 2). This coincident activity enables recruitment of VGCCs to provide local sources of $Ca^{2+}$ able to alter inhibitory synapse strength (Figs. 3–5).

By computationally modelling the effects of inhibitory plasticity at a single neuron level, we predicted that altered inhibition at distinct dendritic compartments dramatically alters pyramidal neuron output (Fig. 6). We confirmed this experimentally showing that action potential generation from proximally and dendritically targeting excitatory inputs is modulated by inhibitory plasticity in corresponding dendritic compartments (Fig. 7).

By expanding our computational model, we show how plasticity at these distinct inhibitory synapses can play roles in the stabilisation of place cell activity within the hippocampus (Fig. 8). This inhibitory plasticity stabilisation ensures place cell fidelity and resilience to interference from activity in multiple different environments.

The mechanisms and network implications of long-term plasticity at glutamatergic synapses have been extensively characterised. However, less attention has been paid to long-term plasticity at inhibitory synapses, which are known to undergo dynamic changes in efficacy[1,27,28]. An array of unique mechanisms discovered for inhibitory plasticity suggests a lack of uniformity across the multiple subtype-specific inhibitory synapses.

Within the hippocampus, synapses from CCK expressing proximally targeting basket cells onto CA1 pyramidal neurons undergo an endocannabinoid-mediated iLTD[36]. Here, mobilisation of retrograde endocannabinoid signalling results in long-term suppression of GABA release[36,62]. We demonstrate that a similar morphological subtype of interneuron, the proximally targeting PV interneurons can also undergo iLTD but in an

endocannabinoid independent mechanism (Fig. 3), highlighting the diversity of plasticity mechanisms even among interneurons with similar morphology[63].

Other long-term inhibitory plasticity mechanisms include a persistent shift in the chloride reversal potential caused by coincident activity-dependent modulation of the KCC2 chloride transporter[39,64,65]. Interestingly, in the hippocampus this iLTD is reported in the feedforward inhibitory pathway for SC innervation of CA1, commensurate with PV basket cell innervation, and is dependent on L-type VGCC and NMDA receptor activation. An alternative set of mechanisms for PV synapse plasticity is reported in the auditory cortex where PV synapses undergo bidirectional iSTDP via BDNF- and $GABA_B$-dependent mechanisms[38]. However, these mechanisms do not appear to apply to PV-iLTD in the hippocampus and moreover, we found no evidence for PV-iLTP. The lack of PV-iLTP is also reported in the prefrontal cortex where SST but not PV synapses undergo iLTP[41]. This suggests that inhibitory plasticity rules may not be conserved across brain regions or that PV synapses undergo multiple forms of inhibitory plasticity. In contrast, the mechanism for iLTP at SST synapses may be broadly conserved, at least between prefrontal cortex and hippocampus, where activation of CAMKII via $Ca^{2+}$ influx to dendrites is found to induce iLTP[41]. The mechanistic differences here relate to the source of $Ca^{2+}$ influx, arising from NMDA receptors in prefrontal cortex and L- and T-type VGCCs in hippocampus. We further show that hippocampal SST synapses can be depressed by non-coincident pre- and postsynaptic spike timing and it will be interesting to find out if this is also the case for synapses in prefrontal cortex.

We show that plasticity at both PV and SST synapses exhibits several key properties: (1) it depends on the coincident activity of inhibitory synapses and postsynaptic action potentials, (2) it is synapse specific, and (3) it relies on postsynaptic $Ca^{2+}$ signalling. These properties are apparently contradictory since synapse-specific inhibition is hyperpolarising, which usually inhibits $Ca^{2+}$ influx and signalling. We show that this apparent contradiction is resolved by recruitment of T-type VGCCs. At resting membrane potentials, T-type VGCCs are in an inactivated state, which can be de-inactivated by a hyperpolarising membrane potential[51,66], this activation profile lends itself perfectly to recruitment by GABAergic synapse activity. Importantly, we show that pairing action potentials before inhibitory input or inhibitory input alone is insufficient to induce inhibitory plasticity. These observations suggest GABA synapse dependent de-inactivation of T-type VGCC is required prior to action potential activation of T-type VGCC, leading to a local synapse-specific source of $Ca^{2+}$ to drive inhibitory plasticity.

Indeed, there is considerable evidence linking GABA signalling and T-type VGCC activation. In the cerebellum and thalamus where T-type VGCCs are widely expressed, T-type VGCCs regulate inhibitory synapse strength[64,67–69]. In the thalamus GABAergic synapses onto thalamocortical neurons de-inactivate T-type VGCCs and reduce inhibitory synaptic strength[68]. This thalamocortical inhibitory plasticity is also dependent on the interaction of calcineurin with $GABA_A$ receptors, similar to findings in the hippocampus[32,52] and those we present here. Interestingly, our data also suggest that under some conditions inhibitory plasticity can engage other VGCCs, for example, L-type VGCCs in SST-iLTP (Fig. 4). This suggests that the contribution of different VGCCs to the intracellular $Ca^{2+}$ transient required for plasticity may vary depending on the precise voltage experienced by a particular synapse.

A striking finding in our results shows that identical induction protocols PV and SST synapses induce opposing forms of plasticity via the differential recruitment of calcineurin and CaMKII. Since CaMKII requires higher $[Ca^{2+}]$ to activate, the differential recruitment of calcineurin and CaMKII could be explained if postsynaptic $[Ca^{2+}]$ is higher at SST versus PV inhibitory synapses. In support of this hypothesis, we show that SST-iLTP relies on L-type as well as T-type VGCCs suggesting a higher level of $Ca^{2+}$ entry. Alternatively, expression levels of VGCCs may increase at more distal dendritic locations and there is evidence that T-type VGCC expression is higher in dendritic regions of pyramidal neurons causing differential regulation of glutamatergic plasticity along the proximal-distal axis of pyramidal neurons[70].

We also show that SST synapses in response to a delayed pre-before postsynaptic pairing protocol undergo LTD similar to that observed at PV synapses. Although not directly tested here, we predict that SST-iLTD may share similar mechanisms to those at PV synapses requiring the recruitment of calcineurin. To draw analogy with the situation at glutamatergic synapses, the spatio-temporal profile of the $Ca^{2+}$ signal in relation to its binding partners dictates if calcineurin or CAMKII pathways are activated[47]. If this is also the case at inhibitory synapses, then we expect that at SST synapses the $Ca^{2+}$ signal in response to longer delays between pre- and postsynaptic activation will be smaller than for coincident activity and therefore preferentially activate calcineurin rather than CAMKII. Conversely, the $Ca^{2+}$ signal in response to coincident pre- and postsynaptic activation or TBS paired with depolarisation will be larger with potentially faster rise times and therefore activate CAMKII rather than calcineurin. The precise complement and timecourse of VGCC activation may be important for dictating the direction of inhibitory plasticity.

Our data support two separate functions of interneuron subtype-specific inhibitory plasticity on hippocampal network function. First, increasing inhibitory inputs to distal dendritic locations on CA1 pyramidal neurons whilst reducing inhibition at proximal locations prioritises response to inputs from CA3 pyramidal neurons via the SC pathway over those from entorhinal neurons via the TA pathway. Second, in our computational model, increasing inhibition at distal dendritic locations inhibits the induction of synaptic plasticity at excitatory synapses[44,59] thereby reducing adaptability of hippocampal representations.

Interneuron-specific inhibitory plasticity at proximal and distal dendritic locations coupled with the anatomical arrangement of SC inputs to proximal dendrites and TA inputs to distal dendrites intuitively predicts that inhibitory plasticity will rebalance the weighting of excitatory inputs in favour of SCs. We formalised these predictions using computational modelling of the CA1 network and then tested them experimentally. We confirmed that PV-iLTD increases CA1 pyramidal neuron responses to SC stimulation, whereas SST-iLTP decreases responses to TA stimulation. Our combination of computational modelling and experimental approaches also showed that the majority of PV interneurons activated by our optogenetic approach are feedforward in the SC pathway but not the TA. Furthermore, our data indicate a limited feedback role for the PV interneurons we activate since PV-iLTD did not impact CA1 pyramidal neuron spike output in response to TA input. This broadly corresponds to anatomical and functional data for PV interneurons in the hippocampus[7,9,43,56]. In contrast, the SST interneurons we stimulate are distally targeting, receive almost no feedforward input and are driven by feedback input from CA1 pyramidal neurons and therefore have all the hallmarks of OLM cells[9,43,57,58].

The implications of reprioritising CA3 input over entorhinal input to CA1 are not straightforward, but parallels can be drawn with the short-term reprioritisation caused by neuromodulator or thalamocortical inputs in cortical circuits[59,71–73]. Often these mechanisms also involve reconfiguration of inhibitory interneuron circuits which are proposed to prioritise input of new sensory information over internal representation on short

timescales including theta cycle timescales[74,75]. The long-term inhibitory plasticity described here is predicted to achieve the reverse outcome prioritising previously learnt associations and making the network less receptive to new information.

Such a scenario would fit with the second role of inhibitory plasticity inhibiting excitatory plasticity and therefore the formation of new representations. SST inhibitory input regulates dendritic excitability and therefore NMDA receptor activation and excitatory synaptic plasticity[15]. Long-term plasticity of SST synapses will change the ability for CA1 pyramidal neurons to undergo induction of excitatory LTP[44,59]. We show that this has major implications for the stability and flexibility of place cells since their formation and remapping depends on excitatory synaptic plasticity driven by dendritic spikes generated by coincident SC and TA inputs[19,61,76]. SST-iLTP prevents place cell representations in environment A being disrupted by different representations in environment B and indeed reduces the ability for place cells to be active in multiple environments. Interestingly, short-term changes in PV and SST interneuron firing rates in response to novel environments may provide a countermechanism to enable new place fields to be formed in novel environments[25]. Our data and modelling therefore provide a mechanism to reconcile the observed stability of place cells across time and their ability to remap in distinct environments[22].

In summary, our data reveal a novel form of inhibitory plasticity in the hippocampus induced by physiological patterns of firing. It has major implications for hippocampal function controlling input-output relationships in CA1 and provides a mechanism to explain a long-standing conundrum regarding place cell stability versus flexibility.

## Methods

**Animal strains and breeding**. All procedures and techniques were conducted in accordance to the UK animals scientific procedures act, 1986 with approval of the University of Bristol ethics committee. To express ChR2 within either PV, SST or Chrna2 expressing interneurons, C57/Bl6 homozygous Ai32 mice (Gt(ROSA) 26Sor$^{tm32(CAG-COP4*H134R/EYFP)Hze}$ Jax Stock number: 024109) were bred with either homozygous PV-Cre (Pvalb$^{tm1(cre)Arbr/J}$ Jax stock number: 017320), SST-Cre (Sst$^{tm2.1(cre)Zjh/J}$ Jax stock number: 013044) or Chrna2-cre$^{44}$ mice creating heterozygous offspring with interneuron-specific expression of ChR2. Mice were group housed on a standard 12 h light/dark cycle (lights on at 7 a.m.) with a controlled average ambient temperature of 21 °C and 45% humidity. For brain slice electrophysiology both male and female mice were used.

**Brain slice preparation**. Brain slices were prepared from 4 to 9-week-old mice following cervical dislocation and decapitation and brains removed and dissected in ice cold cutting solution containing in mM: 205 Sucrose, 10 Glucose, 26 NaHCO$_3$, 2.5 KCl, 1.25 NaH$_2$PO$_4$, 0.5 CaCl$_2$, 5 MgSO$_4$, constantly bubbled with 95% O$_2$ and 5% CO$_2$. Horizontal brain slices, 400 μM thick containing the hippocampus, were prepared via a vibratome (Leica LS1200). Brain slices were transferred to ACSF containing in mM: 124 NaCl, 3 KCl, 24 NaHCO$_3$, 1.25 NaH$_2$PO$_4$ 10 Glucose, 2.5 CaCl$_2$, 1.3 MgSO$_4$, constantly bubbled with 95% O$_2$ and 5% CO$_2$. and incubated at 35 °C for 30 min before being stored at room temperature for at least 30 min before experimentation.

**Whole-cell patch-clamp recordings**. Brain slices were transferred to a submerged slice recording chamber with a constant 2.5 ml/min flow of ACSF (see above), held at 32 °C. Inhibitory plasticity experiments were recorded in the presence of DAP5 (50 μM) and NBQX (20 μM) to isolate GABAergic events. Slices were visualised using infrared DIC optics using a Scientifica SliceScope microscope. Patch electrodes with a resistance of 3–6 MΩ were pulled from borosilicate glass capillaries using a horizontal puller (P-97, Sutter-instruments) and filled with internal solution. For whole-cell voltage-clamp recordings where neurons were held at 0 mV internal solution consisted of in mM: 130 Cs-MeSO$_4$, 4 NaCl, 10 HEPES, 0.5 EGTA, 10 TEA-Cl, 2 Mg-ATP, 0.5 Na2-GTP, 1 QX-314.Cl, adjusted to pH 7.3 with CsOH and ~ 290 mOsm, Cl$^-$ reversal potential −57 mV. For iSTDP experiments an intracellular solution consisting of in mM: 140 K-gluconate, 5 NaCl, 1 MgCl$_2$, 10 HEPES, 4 Mg-ATP, 0.3 Na$_2$-GTP, 0.2 EGTA adjusted to pH 7.3 with KOH and ~ 290 mOsm, Cl$^-$ reversal potential −77 mV. For current-clamp recordings the intracellular solution consisted of in mM: 130 K-gluconate, 8 NaCl, 1 MgCl$_2$, 10 HEPES, 4 Mg-ATP, 0.3 Na$_2$-GTP, 0.2 EGTA adjusted to pH 7.3 with KOH and ~ 290 mOsm, Cl$^-$ reversal −67 mV. For depolarising IPSPs a high [Cl$^-$] internal

solution was used which consisted of in mM: 90 K-gluconate, 40 KCL, 8 NaCl, 1 MgCl$_2$, 10 HEPES, 4 Mg-ATP, 0.3 Na$_2$-GTP, 0.2 EGTA adjusted to pH 7.3 with KOH and ~ 290 mOsm, Cl$^-$ reversal −25 mV. In all experiments a junction potential of ~−15 mV was not compensated.

Recordings of CA1 pyramidal neurons were conducted via a Multiclamp 700A amplifier (Molecular devices), filtered at 6 kHz and digitised at a sampling frequency of 20 kHz using a Micro 1401 data acquisition board (CED). Data were acquired using Signal5 software (CED) and data analysed using custom MATLAB Scripts.

**Synaptic stimulation and plasticity protocols**. For inhibitory plasticity experiments, subtype-specific IPSCs were evoked via optical stimulation of ChR2 via a 470 nm LED (Thorlabs) through a ×40 objective lens using 2–5 ms square pulses of light. Control pathway IPSCs were evoked via 100 μs square pulse electrical stimulation delivered via a monopolar stimulating electrode placed in the pyramidal layer or stratum radiatum. For plasticity experiments each pathway was stimulated every 15 s in an interleaved fashion and synapses from each pathway were checked for independence by a paired pulse protocol (Supplementary Fig. 1).

For light-evoked TBS plasticity, neurons were voltage clamped at 0 mV for the duration of the experiment, light-evoked TBS was applied in voltage clamp and consisted of five bursts delivered at 5 Hz, each burst containing four light pulses at 100 Hz with the protocol repeated five times at 0.033 Hz. For inhibitory spike time dependent plasticity experiments, neurons were voltage clamped at −50 mV. Pairing protocol consisted of 100 pairings at 5 Hz in current clamp (neurons maintained at −50 mV), each consisting of presynaptic light stimulation with a burst of action potentials initiated via somatic current injections (2 ms duration, 1 nA amplitude). For all plasticity experiments if the average control pathway IPSC amplitude deviated > 50% or the series resistance deviated > 20% from baseline values, these cells were excluded from analysis.

Spike probability experiments were conducted in current clamp where 10 EPSPs were evoked at 10 Hz via a bipolar stimulating electrode placed in either the SR or SLM layer to stimulate the SC or TA pathway. Stimulation intensity was adjusted to evoked action potentials in roughly half of the EPSP stimulations.

**Immunohistochemistry**. Brains were fixed via cardiac perfusion of phosphate buffered saline (PBS) followed by 4% formaldehyde in PBS. Brains were removed and stored in PFA for 24 h and then transferred to 30% sucrose PBS solution for 2 days. 50-μm-thick slices were then obtained via cryostat sectioning. Slices were incubated in a blocking solution containing 5% donkey serum and 0.2% Triton X-100 for 90 min at room temperature. Slices were then incubated in room temperature overnight in PBS containing 1% donkey serum and either anti-PV (1:10000, Sigma P3088), anti-SST (1:10000 Santa Cruz SC-7819) or anti-GFP (1:1000 LifeTech A11122) antibodies for PV, SST and ChR2 visualisation, respectively. Slices were then washed with PBS and incubated with secondary antibodies, Alexa-594 (1:1000, LifeTech) or Alexa-488 (1:1000, LifeTech) for 2 h at room temperature, before washing with PBS and mounting on microscope slides with 1:1000 DAPI staining. Slices were then visualised, and images were acquired using a widefield fluorescence microscope. Hippocampal layer regions within the CA1 were defined based on DAPI staining and ChR2 mean fluorescence intensity was quantified using ImageJ software.

**In vitro data analysis**. Experimental unit was defined as cell with only one cell recorded per slice. Up to three cells were recorded from each animal with an average of 1.6 cells per animal. Measurements were taken as an average of four responses to obtain a data point per min, averages represent mean ± SEM. Time series data were normalised to the last 5 min of baseline and plasticity was assessed by comparing the average IPSC amplitude 20–30 min after plasticity induction between the control and test pathway. Owing to the within cell control, plasticity data were analysed using a paired two-tailed Student's t-test between the two pathways. The results of these t-tests are also represented as asterisk on summary histograms of the average test pathway plasticity. Significance assigned * if $p < 0.05$, ** if $p < 0.01$ and *** if $p < 0.001$. Power analysis indicated that minimum sample size of $n = 6$ was required to distinguish between presence and absence of plasticity at 95% confidence intervals with 80% power. Data were processed, analysed and presented using Signal (CED) v5.12, Matlab (R2019a) and Graphpad Prism v8.

**Computational modelling**. Neuron model and network structure: We investigate a feedforward network consisted of a single postsynaptic neuron receiving inputs from the TA pathway, SCs and SST and PV interneurons. The postsynaptic neuron is modelled using a two-compartment, rate-based neuron model. The first compartment represents the distal dendrites of CA1 pyramidal cells, receiving excitatory inputs from the TA pathway and inhibitory inputs from SST interneurons. The second compartment represents the perisomatic region of CA1 pyramidal cells, receiving excitatory inputs from SCs and inhibitory inputs from PV interneurons.

The dendritic compartment's activity, $r_{dend}$, is given as follows:

$$\tau_0 \frac{dr_{dend}}{dt} = -r_{dend} + \left[ TA_{input} - w_{SST} r_{SST} \right]_+ ,$$

where $[\cdot]_+$ denotes a rectification that sets negative values to 0, $\tau_0$ is a time constant, $TA_{input}$ is the TA pathway input, $w_{SST}$ is the synaptic weight from SST interneurons to the CA1 pyramidal cell and $r_{SST} = 1$ (Fig. 8) is the SST interneuron activity. In Fig. 6 and Supplementary Fig. 5, $r_{SST} = \alpha\, SC_{input} + 0.5\,\beta\, TA_{input} + 2\,\gamma\, r_{soma}$ and the parameters $\alpha$, $\beta$, and $\gamma$ were set to either 0 or 1 depending on whether we simulated feedforward and feedback inhibition.

The somatic compartment's activity, $r_{soma}$, is given as follows:

$$\tau_0 \frac{dr_{soma}}{dt} = -r_{soma} + g\left(r_{dend} + w_{SC} SC_{input} - w_{PV} r_{PV}\right),$$

where $SC_{input}$ is the activity of SC input neurons, $w_{SC}$ is the synaptic weight from SC inputs, $w_{PV}$ is the synaptic weight from PV interneurons to the CA1 pyramidal cell, $r_{PV} = 1$ (Fig. 8) is the PV interneuron activity. In Fig. 6 and Supplementary Fig. 5, $r_{PV} = 0.2\,\alpha\, SC_{input} + 0.1\,\beta\, TA_{input} + 0.2\,\gamma\, r_{soma}$ and the parameters $\alpha$, $\beta$, and $\gamma$ were set to either 0 or 1 depending on whether we simulated feedforward and feedback inhibition. The dendritic non-linearity $g$ is a non-linear function given as follows:

$$g(x) = \frac{4}{3}\left[\tanh\left(\frac{2x}{5}\right)\right]_+ .$$

**TA and SC inputs**. The simulated CA1 pyramidal cells receive excitatory inputs from the TA pathway and SCs. The input from the TA pathway is simulated as $TA_{input} = \mu_{TA} + \xi_{TA}$, where $\mu_{TA} = 2$ and $\xi_{TA}$ is generated from an Ornstein–Uhlenbeck process with a time constant of 50 ms, mean 0 and variance 0.5. The SC inputs are generated from $N_{SC}$ input neurons and each input neuron is tuned to a specific location such that their firing rates span over the entire environment. All place fields of SC input neurons have the same tuning width $\sigma_{SC}$ and amplitude $A_{SC}$.

For the simulations involving exploration, the simulated animal explores an annular track of length $L$ with speed $v$. The activity of an SC input neuron with place field centred at position $p_0$ is as follows:

$$SC_{input}(p) = A_{SC}\exp\left(-\frac{d^2}{2\sigma_{SC}^2}\right) + \xi_{SC},$$

where $p$ is the animal's position, $d$ is the distance between the $p$ and $p_0$ along the track and $\xi_{SC}$ is generated from an Ornstein–Uhlenbeck process with a time constant of 50 ms, mean 0 and variance 0.5.

In Fig. 6, we simulate an artificial stimulation of TA and SC. Therefore, for these simulations, SC inputs are not spatially tuned. Instead, they are simulated as $SC_{input} = \mu_{SC} + \xi_{SC}$, where $\mu_{SC} = 2$.

**Excitatory plasticity model**. In Fig. 8, synaptic weights from SC input neurons to CA1 pyramidal neurons are plastic. These connections follow a Hebbian-type plasticity rule in which changes in synaptic weights depend on coincident pre- and postsynaptic activity. The postsynaptic term is given by the product of dendritic and somatic activity. This is motivated by recent findings suggesting that place fields are modified and formed following coincident TA and SC inputs[19,61]. The excitatory synaptic weight $w_{ij}$ from input neuron $j$ to postsynaptic neuron $i$ is updated as follows:

$$\frac{dw_{ij}}{dt} = \eta_{SC}\left(r_{dend}^i r_{soma}^i - 0.1\right)_+ r^j,$$

where $\eta_{SC}$ is the learning rate for SC connections, $r^j$ is the presynaptic neuron activity, $r_{soma}^i$ is the somatic activity of the postsynaptic neuron and $r_{dend}^i$ is the dendritic activity of the postsynaptic neuron. Because this rule is inherently unstable, synaptic weights are also normalised as commonly done[77]. After every weight update, we subtract the average synaptic weight $\sum_j w_{ij}/N_{SC}$ from all weights and add a constant term (here 2). Negative weights are then rectified to 0.

**Inhibitory plasticity model**. We implement an inhibitory plasticity rule inspired by our experimental findings. Under a rate-based framework, these plasticity rules are assumed to mirror those found during theta oscillations. Synaptic weights from PV interneurons onto CA1 pyramidal cells follow a rate-based Hebbian plasticity rule in which the coactivation of pre- and postsynaptic neurons leads to LTD:

$$\frac{dw_{PV}}{dt} = -\eta_{PV} r_{PV} r_{soma},$$

where $\eta_{PV}$ is the learning rate for PV connections and $r_{soma}$ is pyramidal cell somatic activity. Synaptic weights from SST interneurons onto CA1 pyramidal cells follow a Hebbian plasticity rule in which the coactivation of pre- and postsynaptic neurons leads to LTP:

$$\frac{dw_{SST}}{dt} = \eta_{SST} r_{SST} r_{soma},$$

where $\eta_{SST}$ is the learning rate for SST connections. Both PV and SST synaptic weights are bounded between $w_{min} = 0$ and $w_{max} = 10$.

## Table 1 Model parameters.

| Name | Value | Description |
|---|---|---|
| **Neuron model** | | |
| $\tau_0$ | 1.25 ms | Time constant for the dynamics |
| **Network parameters** | | |
| $N_{SC}$ | 40 | Number of SC input neurons |
| $N_{SST}$ | 1 | Number of SST interneurons |
| $N_{PV}$ | 1 | Number of PV interneurons |
| **Plasticity model** | | |
| $\eta_{SST}$ | $2.0 \times 10^{-4}$ ms$^{-1}$ | SST plasticity learning rate |
| $\eta_{PV}$ | $2.0 \times 10^{-4}$ ms$^{-1}$ | PV plasticity learning rate |
| $\eta_{SC}$ | $2.5 \times 10^{-5}$ ms$^{-1}$ | Learning rate for Schaffer collaterals |
| **Place-tuned input** | | |
| $A_{SC}$ | 6.0 | Presynaptic place field amplitude |
| $\sigma_{SC}$ | 2.0 | Presynaptic place field width |
| **Simulation parameters** | | |
| $L$ | 50 | Track length |
| $v$ | $2.5 \times 10^{-3}$ ms$^{-1}$ | Animal speed |

**Environment switch**. In Fig. 8, we simulate a feedforward network, while an animal runs through an annular track. When the animal starts exploring environment A for the first time, the initial synaptic weights for the SC inputs are drawn from a lognormal distribution with underlying normal distribution with mean 0 and standard deviation 0.1. The synaptic weights are then multiplied by 0.1 and two neighbouring inputs are randomly chosen and their synaptic weights are set to 0.6. This imposes a small structure in the synaptic weights and ensures that input neurons are able to induce postsynaptic activity. The animal then explores environment A for ten laps. Next, the animal is moved to a novel environment (environment B). We simulate the switch to a novel environment by randomly shuffling the identity of the SC inputs to the CA1 pyramidal neuron. The animal then explores environment B for 15 laps. Subsequently, the animal is moved back to environment A (environment A′), which is implemented by returning the SC inputs to the original identity. Finally, the animal explores environment A′ for another ten laps.

**Parameters and simulations**. All simulations were implemented in python and are available at ModelDB. The parameters used in our simulations can be found in Table 1.

**Reporting summary**. Further information on research design is available in the Nature Research Reporting Summary linked to this article.

## Data availability
Further information and data that support the findings of this study are available upon reasonable request from the corresponding authors C.Clopath@imperial.ac.uk and Jack. Mellor@bristol.ac.uk. Computational modelling data are generated by the simulation code (see "Code availability").

## Code availability
The code for all simulations in this paper is available on ModelDB (Accession Number 259481).

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

## Acknowledgements

We thank Rui Ponte Costa, David Dupret and Andrew Randall for critical input to previous versions of the paper and all members of the Clopath and Mellor groups for discussion. We also thank Klas Kullander for providing Chrna2-cre mice. This work was supported by Biotechnology and Biological Sciences Research Council (BBSRC), BB/ N013956/1, BB/N019008/1, Wellcome Trust 200790/Z/16/Z, 101029/Z/13/Z, Simons Foundation 564408, EPSRC EP/R035806/1.

## Author contributions

Conceptualisation: C.C. and J.R.M.; methodology: M.U., V.P. and S.E.L.C.; investigation: M.U., V.P. and S.E.L.C.; visualisation: M.U. and V.P.; writing—original draft: M.U., V.P., C.C. and J.R.M.; writing—review and editing: M.U., V.P., C.C. and J.R.M.; funding acquisition: C.C. and J.R.M.; supervision: C.C. and J.R.M.

## Competing interests

The authors declare no competing interests.
