## [Peer Review File · Nature Communications]

Reviewers' Comments:

Reviewer #1:

Remarks to the Author:

Udakis et al. describe interneuron-specific forms of inhibitory synaptic plasticity in hippocampus, which contributes to modulation of the excitation of CA1 pyramidal cells. In their paradigm, TBS stimulation of PV or SST axons induces long-term depression of the PV synaptic inputs, and long-term potentiation of SST synaptic inputs, respectively. PV-iLTD and SST-iLTP can also be induced using STDP pairing, with strong effects at 0ms delays. Using pharmacology, the authors determine that the iLTD is dependent on T-type calcium channels and a calcineurin-mediated pathway, and iLTP on both T-type and L-type calcium channels and a CaMKII-dependent pathway. Based on previously described innervation patterns suggesting that PV cells predominantly inhibit perisomatic dendrites (through feed forward inhibition) and SST cells distal dendrites (through feedback inhibition), the authors derive a multicompartamental model for plasticity in which the changes in PV and SST interneuron inputs modulate Schaffer collateral and temporammonic pathway-mediated activity of pyramidal cells, respectively. Further modeling experiments then clearly lend support to the hypothesis that these forms of plasticity explain the stability of place cell activity.

This paper describes a novel form of differential inhibitory synaptic plasticity in hippocampus with strong implications (and a possible explanation) for place cell plasticity and stability. I consider these important new insights and support publication. The model provides a nice framework for further experimental testing.

I have two points that I think need to be addressed before publication.

Major points:

1. I find it difficult to comprehend what is being stimulated in the control pathway. The authors mention that they are recording IPSCs upon electrical stimulation of inputs in the pyramidal cell layer or stratum radiatum. But isn't a large fraction of the inhibitory inputs in these regions exactly derived from the inputs they are testing? So, I find it a bit puzzling that the consecutive E-L and L-E pulse experiment in Suppl. Fig 1h remains unaffected. I don't doubt the outcome of the experiments, but perhaps the authors could comment on why they think this is the case. Does it have to do with a limited number of inputs that are labeled with channelrhodopsin vs what is recruited through electrical stimulation (i.e. the monopolar stimulating electrode could cause wide spread activation of synapses – even beyond the targeted pathways).
2. It is also not clear to me how the TBS results (Fig 1) link to the STDP results (Fig 2). What would be the induction mechanism in the TBS experiments, in which the cells are held at 0mV and glutamate receptors are blocked? Do the authors think that the mechanisms are similar to the STDP paradigm? I find this somewhat puzzling, as the authors state that iSTDP depended on T-type calcium channels, which require hyperpolarization for de-inactivation. Would this happen in a neuron held at 0 mV?

Minor points.

3. Out of curiosity. Do the authors think that the -60ms pairing-induced SST-iLTD depends on the same mechanism as PV-iLTD; and would this be similar to the PV-iLTD at 0ms?
4. Weren't the authors surprised that SST-iLTP required the activation of CAMKII, which normally requires a strong depolarization? Do bAPs travel far enough into the distal dendrites to cause such a strong depolarization? Perhaps the authors could elaborate a bit on this in the discussion section.

5. It would be helpful to indicate in the graphs over which periods the data were averaged to plot the 'before' and 'after' plasticity-histograms.

Reviewer #2:

Remarks to the Author:

The manuscript by Udakis et al explores the induction requirements and possible circuit consequences of plasticity at interneuron synapses on pyramidal neurons in hippocampal CA1. The study ranges widely, from optogenetic and pharmacological manipulations in vitro to a computational model of how plasticity of feed-forward inhibition of temporoammonic and Schaffer collateral inputs could contribute to stabilizing place cells. The study is interesting although there are gaps in the evidence linking the different parts of the work.

1. Fig. 1 shows convincing and abrupt changes in IPSC amplitude in opposite directions following optogenetic TBS, but the rest of the data using spike-timing protocols show variable and slow changes in IPSCs. Why?
2. It is curious that the control pathway went in opposite directions in the experiments reported in Fig. 1.
3. The authors apply repeated t-tests when comparing various manipulations (pharmacology, different timing protocols), and do not take into account multiple comparisons. An ANOVA would be more appropriate, although the experiments may have been underpowered.
4. The authors use absence of significance to argue for no effect, which is not correct.
5. The involvement of T-type channels is inferred on the basis of pharmacology, but the authors do not show how the different protocols may have led to different degrees of Ca²⁺ influx via these channels, either with membrane voltage or intracellular Ca²⁺ recordings or simulations.
6. The near-threshold action potential trains reported in Fig. 7 are interesting and the effects of iLTP induction at PV vs SST synapses are striking. However, I am very puzzled how the authors obtained trains of spikes when stimulating the temporoammonic input. Several previous papers report that this only leads to very large IPSPs because feed-forward inhibition swamps any direct excitation of the pyramidal neuron dendrites.
7. Given the profound short-term depression seen at PV interneuron synapses, I would expect to see markedly different effects of iLTD on early vs late spikes in the 10-spike trains.
8. The authors do not actually show that i-LTP/i-LTD has been induced in these experiments. This should be possible. Indeed, it might even be possible to show how optogenetic induction of plasticity alters disynaptic IPSPs evoked by the electrical stimuli.

Reviewer #1 (Remarks to the Author):

Udakis et al. describe interneuron-specific forms of inhibitory synaptic plasticity in hippocampus, which contributes to modulation of the excitation of CA1 pyramidal cells. In their paradigm, TBS stimulation of PV or SST axons induces long-term depression of the PV synaptic inputs, and long-term potentiation of SST synaptic inputs, respectively. PV-iLTD and SST-iLTP can also be induced using STDP pairing, with strong effects at 0ms delays. Using pharmacology, the authors determine that the iLTD is dependent on T-type calcium channels and a calcineurin-mediated pathway, and iLTP on both T-type and L-type calcium channels and a CaMKII-dependent pathway. Based on previously described innervation patterns suggesting that PV cells predominantly inhibit perisomatic dendrites (through feed forward inhibition) and SST cells distal dendrites (through feedback inhibition), the authors derive a multicompartmental model for plasticity in which the changes in PV and SST interneuron inputs modulate Schaffer collateral and temporammonic pathway-mediated activity of pyramidal cells, respectively. Further modeling experiments then clearly lend support to the hypothesis that these forms of plasticity explain the stability of place cell activity.

This paper describes a novel form of differential inhibitory synaptic plasticity in hippocampus with strong implications (and a possible explanation) for place cell plasticity and stability. I consider these important new insights and support publication. The model provides a nice framework for further experimental testing.

I have two points that I think need to be addressed before publication.

Major points

1. *I find it difficult to comprehend what is being stimulated in the control pathway. The authors mention that they are recording IPSCs upon electrical stimulation of inputs in the pyramidal cell layer or stratum radiatum. But isn't a large fraction of the inhibitory inputs in these regions exactly derived from the inputs they are testing? So, I find it a bit puzzling that the consecutive E-L and L-E pulse experiment in Suppl. Fig 1h remains unaffected. I don't doubt the outcome of the experiments, but perhaps the authors could comment on why they think this is the case. Does it have to do with a limited number of inputs that are labeled with channelrhodopsin vs what is recruited through electrical stimulation (i.e. the monopolar stimulating electrode could cause wide spread activation of synapses – even beyond the targeted pathways).*

We have evidence showing that electrical stimulation does recruit a more diverse and heterogeneous population of interneurons than optogenetic stimulation, as the reviewer suggests. A comparison of the decay kinetics for electrical and optogenetic stimulation reveals that electrical stimulation of the pyramidal layer produces slower IPSCs than PV optogenetic stimulation and electrical stimulation of Stratum Radiatum produces faster IPSCs than SST optogenetic stimulation (see new data in Supplementary Figure 1). These data highlight that different (but potentially partially overlapping) populations of interneurons are engaged by electrical and optogenetic stimulation as confirmed by the functional independence tests shown in Supplementary Figure 1h. On reflection this is perhaps not so surprising since the axonal and dendritic processes of many different interneurons are found in multiple layers of CA1 and will therefore be activated by electrical stimulation. For example, electrical stimulation in the pyramidal layer would likely activate populations of PV and CCK expressing basket cells but also the axons of interneurons such as OLM or

bistratified interneurons whose axons pass through the pyramidal layer to target synapses in more distal dendritic regions. So electrical stimulation will activate multiple diverse interneurons whereas optogenetic stimulation only activates PV expressing interneurons.

2. *It is also not clear to me how the TBS results (Fig 1) link to the STDP results (Fig 2). What would be the induction mechanism in the TBS experiments, in which the cells are held at 0mV and glutamate receptors are blocked? Do the authors think that the mechanisms are similar to the STDP paradigm? I find this somewhat puzzling, as the authors state that iSTDP depended on T-type calcium channels, which require hyperpolarization for de-inactivation. Would this happen in a neuron held at 0 mV?*

This is an excellent point (related to the 2nd reviewer's point 5) and we have addressed it with new experiments to test the role of T-type VGCCs in TBS induced SST-iLTP (Supplementary Figure 4d). These data show that the T-type antagonist ML218 completely blocks the plasticity indicating that both TBS and STDP induction protocols engage similar mechanisms. As the reviewer points out, it is initially surprising that T-type VGCCs could be engaged when the cell is voltage clamped at 0mV. However, voltage clamp of dendrites is difficult and the best estimates suggest that only the somatic membrane can be properly controlled under conditions of synaptic stimulation (Williams and Mitchell, 2008). Therefore, the local dendritic membrane potential at stimulated inhibitory synapses during TBS will likely be close to the Cl⁻ reversal potential – despite voltage clamp at 0mV – allowing the de-inactivation of T-type VGCCs.

Interestingly, our data also suggest that under some conditions inhibitory plasticity can engage other VGCCs, for example L-type VGCCs in SST-iLTP (Figure 4). This suggests that the contribution of different VGCCs to the intracellular Ca²⁺ transient required for plasticity may vary depending on the precise voltage experienced by a particular synapse. We have focussed on the synaptic activity patterns most likely relevant to ongoing hippocampal activity for which the T-type VGCCs play a key role but this does not mean that other VGCCs may be important during other stimulation paradigms. We now include these points in the discussion.

Minor points

3. *Out of curiosity. Do the authors think that the -60ms pairing-induced SST-iLTD depends on the same mechanism as PV-iLTD; and would this be similar to the PV-iLTD at 0ms?*

Although we have not explored this possibility within the manuscript, we would predict that the two forms of LTD at PV and SST synapses share at least some components of a common mechanism. Our data show that a Ca²⁺ signal via VGCCs is required for PV-iLTD and SST-iLTP and that if this activates calcineurin then LTD occurs at PV synapses whereas if CAMKII is activated then LTP occurs at SST synapses. To draw analogy with the situation at glutamatergic synapses, the spatiotemporal profile of the Ca²⁺ signal in relation to its binding partners dictates which signalling pathway is activated. If this is also the case at inhibitory synapses, then we expect that at SST synapses the Ca²⁺ signal in response to longer delays between pre- and post-synaptic activation will be smaller than for coincident activity and therefore preferentially activate calcineurin rather than CAMKII. As described in the previous point, it may be that different complements of VGCCs are activated by the different plasticity inducing protocols and they may each couple differentially to calcineurin and CAMKII. This could lead to a divergence between the mechanisms at PV and SST synapses but fundamentally

plasticity at each synapse is dictated by the spatiotemporal Ca^{2+} profile. These ideas are now included within the discussion.

4. *Weren't the authors surprised that SST-iLTP required the activation of CAMKII, which normally requires a strong depolarization? Do bAPs travel far enough into the distal dendrites to cause such a strong depolarization? Perhaps the authors could elaborate a bit on this in the discussion section.*

We have previously modelled the activation of CAMKII in dendrites and shown that it depends not so much on the level of depolarisation but rather the combination of fast and slow Ca^{2+} signals to bind to the high and low affinity sites (Griffith et al., 2016). In many situations this translates practically into large depolarisations that provide high Ca^{2+} concentrations but it can also require Ca^{2+} signals from different sources (eg (Tigaret et al., 2016)). Our induction protocols in general require considerable postsynaptic depolarisation – either in the form of continuous depolarisation or high frequency firing of multiple action potentials (single action potentials are not sufficient). Our previous data measuring Ca^{2+} signals in dendrites suggests that bursts of action potentials can generate substantial Ca^{2+} increases necessary for plasticity at glutamatergic synapses (Tigaret et al., 2016; Tigaret et al., 2018). We do not know the spatiotemporal profile of Ca^{2+} signals at inhibitory synapses during plasticity induction but it is entirely plausible that it fulfils the criteria for CAMKII activation. Indeed, other groups have shown that distal inhibitory synapses are also potentiated via NMDA receptor mediated activation of CAMKII (eg. (Chiu et al., 2018)). We now elaborate on these points in the discussion.

5. It would be helpful to indicate in the graphs over which periods the data were averaged to plot the 'before' and 'after' plasticity-histograms.

The comparison histograms represent an average of the IPSC amplitudes for the 10 minute period 20-30 minutes after plasticity induction comparing the test and control pathway. This is made clear in the methods section. We prefer not to indicate this on the figures themselves to avoid cluttering the data.

Reviewer #2 (Remarks to the Author):

The manuscript by Udakis et al explores the induction requirements and possible circuit consequences of plasticity at interneuron synapses on pyramidal neurons in hippocampal CA1. The study ranges widely, from optogenetic and pharmacological manipulations in vitro to a computational model of how plasticity of feed-forward inhibition of temporoammonic and Schaffer collateral inputs could contribute to stabilizing place cells. The study is interesting although there are gaps in the evidence linking the different parts of the work.

1. *Fig. 1 shows convincing and abrupt changes in IPSC amplitude in opposite directions following optogenetic TBS, but the rest of the data using spike-timing protocols show variable and slow changes in IPSCs. Why?*

The slow development of plasticity evoked by STDP protocols is a consistent observation for plasticity at glutamatergic and inhibitory synapses (for examples see (Bi and Poo, 1998; Froemke and Dan, 2002; Woodin et al., 2003; Campanac and Debanne, 2008; Tigaret et al., 2016; Tigaret et al., 2018; Vickers et al., 2018)). This contrasts with the much faster evolution of plasticity induced by high frequency stimulation (eg (Buchanan and Mellor, 2007)). It is not clear why this difference exists and to our knowledge no-one has studied this definitively. One possibility is that high frequency stimulation or chronic depolarisation induces a short-term potentiation (or depression) that creates the impression of rapidly evolving plasticity but which in fact simply sits on top of the slower development of long-term potentiation (or depression) (see for example (Hoffman et al., 2002)). In such a scenario rapid trafficking of GluA1 containing glutamate receptors into synapses can cause a short-term increase in synaptic strength. These receptors are then replaced on a longer-term basis by GluA1 and GluA2 containing receptors. However, it is not clear why or how STDP might bypass the initial GluA1 trafficking event and we are not aware of any studies that directly address this question.

In summary, the observation that plasticity evolves slowly for STDP is common to multiple studies in different groups and therefore a general question for the field which requires further targeted investigation.

2. *It is curious that the control pathway went in opposite directions in the experiments reported in Fig. 1*

The control pathway data for each of these experiments show inherent variability whereby the control pathway can increase or decrease over time. This is random and expected and the data for all control pathways confirms a normal distribution with mean of ~100% (see Figure 1 below). Analysis of these data reveal that the average grouped data control pathways for PV-iLTD (or SST-iLTP) can either increase or decrease regardless of whether there is plasticity in the test pathway (see Figures 1, 2, 3 & 4). Individual experiments show a normally distributed spread of control pathway changes but in none of these cases does the average of the control pathway show a significant change from baseline. On average, the control pathways for PV experiments are $102.3 \pm 2.2\%$ at the end of experiments and for SST experiments they are $101.4 \pm 2.0\%$. We applied an exclusion criterion of >50% increase or decrease in control pathway for all our experiments but this resulted in very few exclusions as can be deduced from the distributions shown in Figure 1 below. This exclusion criterion is now included in the methods. In summary, we do not observe a consistent plasticity of control pathways.

Electrical Stim Control Pathway.

	PV Ctr Path	SST Ctr Path
Number of values	75	81
25% Percentile	89.45	87.98
Median	102.2	99.03
75% Percentile	115.9	115.1
Mean	102.3	101.4
Std. Deviation	19.30	17.85
Std. Error of Mean	2.229	1.984

The inclusion of control pathways is an important component of our experimental design since it allows within experiment comparisons and therefore paired statistical tests to assess the induction of synaptic plasticity. This increases the power of the conclusions drawn and it is important to highlight that the within cell comparison of the control and test pathways show consistent decreases between pathways for PV-iLTD and consistent increases for SST-iLTP.

3. The authors apply repeated t-tests when comparing various manipulations (pharmacology, different timing protocols), and do not take into account multiple comparisons. An ANOVA would be more appropriate, although the experiments may have been underpowered.

This is an important point that requires clarification of our statistical approach. The design of our plasticity experiments which include a control pathway and a test pathway ensures that we are able to do within experiment statistical comparisons using paired t-tests between control and test pathways, as displayed in individual histograms (Figures 1, 2, 3, 4). We use these statistical comparisons to make conclusions about the data. We also include a summary histogram in which we perform multiple one sample t-tests. We agree with the reviewer that the use of multiple t-tests in this histogram and the use of ns to signify no effect is inaccurate. To address this, we have removed any use of multiple one sample t-tests for the summary histograms and indication of non-significance. Instead, in our summary histogram we now use an asterisk to signify significance based on the paired t-tests of the individual plasticity histograms. This is now clearly stated in each of the figure legends.

4. The authors use absence of significance to argue for no effect, which is not correct.

Our power analysis using effect sizes for plasticity experiments calculated from our data indicates that sample size $n = 6$ is required for 80% power at 95% confidence intervals. Therefore, we can be 80% confident that we are not assigning “no effect” incorrectly based on our statistical approach using within experiment control pathways. This analysis is based on the detection of “all or nothing” plasticity and not a graded response so we are not able to make definitive statements about lack of smaller effects. We now include a description of statistical power in the methods to clarify our statements based on the statistical approach.

5. The involvement of T-type channels is inferred on the basis of pharmacology, but the authors do not show how the different protocols may have led to different degrees of Ca²⁺ influx via these channels, either with membrane voltage or intracellular Ca²⁺ recordings or simulations.

This is an interesting point. The involvement of T-type VGCCs depends on specific membrane voltage conditions (also raised by reviewer 1, point 2) and we have now tested these with new experiments (Supplementary Figure 4). We propose that T-type VGCCs require de-inactivation by inhibitory synaptic input to then activate with a subsequent burst of back-propagating action potentials. To test this, we switched inhibitory synaptic inputs to be depolarising rather than hyperpolarising by increasing the internal Cl⁻ concentration from 7mM to 50mM. We then tested SST synapses with coincident pre- and post-synaptic STDP activation that normally induces LTP. Increasing internal Cl⁻ concentration prevented LTP induction (Supplementary Figure 4C) supporting the conclusion that hyperpolarisation and therefore de-inactivation of T-type VGCCs is necessary for inhibitory plasticity.

As discussed above in response to reviewer 1 point 2, the new data showing a role for T-type VGCCs in plasticity induced whilst holding the membrane potential at 0mV is initially counter-intuitive (Supplementary Figure 4D). However, on reflection the poor voltage clamp achieved in neurons, particularly of synaptic voltages in dendrites, means that significant hyperpolarisation will still occur despite holding the cell soma at 0mV.

6. The near-threshold action potential trains reported in Fig. 7 are interesting and the effects of iLTP induction at PV vs SST synapses are striking. However, I am very puzzled how the authors obtained trains of spikes when stimulating the temporoammonic input. Several previous papers report that this only leads to very large IPSPs because feed-forward inhibition swamps any direct excitation of the pyramidal neuron dendrites.

We believe the answer to this comment lies principally in the frequency of stimulation used for the temporoammonic input. The reviewer points out that several prior studies (eg (Dvorak-Carbone and Schuman, 1999; Jarsky et al., 2005; Sun et al., 2014; Milstein et al., 2015; Masurkar et al., 2017)) found difficulty in generating spikes in CA1 pyramidal neurons by stimulating temporoammonic inputs. These studies mostly used either single stimuli or trains of high frequency stimuli (50-100 Hz). Our data show that we almost never see spikes in response to single stimuli (Supplementary Figure 6). It has also been shown that the excitatory-inhibitory balance for temporoammonic synaptic inputs decreases dramatically at stimulation frequencies ≥ 20 Hz due to recruitment of inhibition (Booth et al., 2014). We use a stimulation frequency of 10 Hz that does not engage such strong inhibition whilst still causing prominent facilitation of the excitatory synaptic inputs. Therefore, our data do agree with the previous reports, but we chose our stimulation frequency carefully to enable us to study the spike output in response to temporoammonic stimulation.

We certainly find that spikes are harder to generate in the temporoammonic pathway than Schaffer collateral as might be expected for synaptic inputs that occur in the more distal dendritic regions but our data clearly show that it is possible to generate spikes in response to temporoammonic input. The fact that we see differences between the ability to generate spikes in temporoammonic and Schaffer collateral pathways supports other data from our group using pharmacology to show we are able to selectively stimulate these pathways (Palacios-Filardo et al 2020 BioRxiv, Supplementary Figure 1).

7. Given the profound short-term depression seen at PV interneuron synapses, I would expect to see markedly different effects of iLTD on early vs late spikes in the 10-spike trains.

This is an excellent suggestion and we have now performed additional analyses to investigate the effects of inhibitory plasticity on early vs late spikes (Supplementary Figure 6). We agree frequency-dependent short-term depression at inhibitory synapses is predicted to enhance the spike probability in response to late stimuli in the train of spikes. The analysis revealed that this was indeed the case. Spike probability in response to stimulation of either SC or TA pathways was initially low and increased substantially towards the later stimuli within the train (Supplementary Figure 6), presumably due to a combination of depression in the di-synaptic feedforward inhibition and facilitation of excitatory inputs. In the SC pathway we assume that a significant proportion of the feedforward inhibition is mediated by PV interneurons and although these synapses show short-term depression a residual level of inhibition is still present in response to late stimuli in the train. The question is then whether PV-iLTD has a greater effect on spike probability earlier in the train due to there being more PV inhibition? This will depend on whether the magnitude of PV-iLTD is consistent regardless of prior synaptic activity and the amount of presynaptic release. Our analysis shows an equal effect of PV-iLTD on the spike probability for early and late stimuli within the train (Supplementary Figure 6b) suggesting that PV-iLTD is postsynaptically expressed and not dependent on presynaptic release. It also indicates that PV feedforward interneurons are recruited equally reliably across the train. Conversely, SST synapses, which are typically feedback interneurons, would be expected to be preferentially recruited at later phases of the train after pyramidal cell spiking has occurred. We find this is indeed the case where SST-iLTP reduces the spike probability in response to later stimuli much more reliably than for early stimuli (Supplementary Figure 6c). These data and ideas are now included in the results section.

8. The authors do not actually show that i-LTP/i-LTD has been induced in these experiments. This should be possible. Indeed, it might even be possible to show how optogenetic induction of plasticity alters disynaptic IPSPs evoked by the electrical stimuli.

We consistently show that coincident pre- and post-synaptic activation of inhibitory synapses induces inhibitory synaptic plasticity at these synapses (Figures 2, 3, 4) and we use the exact same protocols in Figure 7 to assess the impact of plasticity on CA1 spike output. Therefore, we do not think it is critical to demonstrate that plasticity is induced within these experiments. We have looked at our data to see if it is possible to measure the inhibitory component of the membrane voltage trace during the trains of stimuli but we are not confident this will reveal anything meaningful since the membrane voltage is composed of multiple components including EPSPs, IPSPs, action potentials and afterhyperpolarisations. Thus, specifically measuring PV or SST mediated disynaptic IPSPs is not feasible within our current dataset.

Demonstrating that inhibitory plasticity alters disynaptic inhibition would certainly be an interesting experiment to perform. However, it would require separate experiments without the spike analysis, using voltage clamp to isolate IPSCs by holding cells at the reversal potential for glutamatergic synaptic transmission. These experiments would necessarily require very different conditions and therefore detract from the relevance to the spike probability experiments. Therefore, although interesting, we do not believe they are necessary for the present manuscript.

References

- Bi GQ, Poo MM (1998) Synaptic modifications in cultured hippocampal neurons: Dependence on spike timing, synaptic strength, and postsynaptic cell type. *Journal of Neuroscience* 18:10464-10472.
- Booth CA, Brown JT, Randall AD (2014) Neurophysiological modification of CA1 pyramidal neurons in a transgenic mouse expressing a truncated form of disrupted-in-schizophrenia 1. *Eur J Neurosci* 39:1074-1090.
- Buchanan KA, Mellor JR (2007) The development of synaptic plasticity induction rules and the requirement for postsynaptic spikes in rat hippocampal CA1 pyramidal neurones. *J Physiol* 585:429-445.
- Campanac E, Debanne D (2008) Spike timing-dependent plasticity: a learning rule for dendritic integration in rat CA1 pyramidal neurons. *J Physiol* 586:779-793.
- Chiu CQ, Martenson JS, Yamazaki M, Natsume R, Sakimura K, Tomita S, Tavalin SJ, Higley MJ (2018) Input-Specific NMDAR-Dependent Potentiation of Dendritic GABAergic Inhibition. *Neuron* 97:368-377 e363.
- Dvorak-Carbone H, Schuman EM (1999) Long-term depression of temporoammonic-CA1 hippocampal synaptic transmission. *J Neurophysiol* 81:1036-1044.
- Froemke RC, Dan Y (2002) Spike-timing-dependent synaptic modification induced by natural spike trains. *Nature* 416:433-438.
- Griffith T, Tsaneva-Atanasova K, Mellor JR (2016) Control of Ca²⁺ Influx and Calmodulin Activation by SK-Channels in Dendritic Spines. *PLoS computational biology* 12:e1004949.
- Hoffman DA, Sprengel R, Sakmann B (2002) Molecular dissection of hippocampal theta-burst pairing potentiation. *P Natl Acad Sci USA* 99:7740-7745.
- Jarsky T, Roxin A, Kath WL, Spruston N (2005) Conditional dendritic spike propagation following distal synaptic activation of hippocampal CA1 pyramidal neurons. *Nat Neurosci* 8:1667-1676.
- Masurkar AV, Srinivas KV, Brann DH, Warren R, Lowes DC, Siegelbaum SA (2017) Medial and Lateral Entorhinal Cortex Differentially Excite Deep versus Superficial CA1 Pyramidal Neurons. *Cell reports* 18:148-160.
- Milstein AD, Bloss EB, Apostolides PF, Vaidya SP, Dilly GA, Zemelman BV, Magee JC (2015) Inhibitory Gating of Input Comparison in the CA1 Microcircuit. *Neuron* 87:1274-1289.
- Sun Y, Nguyen AQ, Nguyen JP, Le L, Saur D, Choi J, Callaway EM, Xu X (2014) Cell-type-specific circuit connectivity of hippocampal CA1 revealed through Cre-dependent rabies tracing. *Cell reports* 7:269-280.
- Tigaret CM, Olivo V, Sadowski JH, Ashby MC, Mellor JR (2016) Coordinated activation of distinct Ca(2+) sources and metabotropic glutamate receptors encodes Hebbian synaptic plasticity. *Nature communications* 7:10289.
- Tigaret CM, Chamberlain SEL, Sadowski J, Hall J, Ashby MC, Mellor JR (2018) Convergent Metabotropic Signaling Pathways Inhibit SK Channels to Promote Synaptic Plasticity in the Hippocampus. *J Neurosci* 38:9252-9262.
- Vickers ED, Clark C, Osypenko D, Fratzl A, Kochubey O, Bettler B, Schneggenburger R (2018) Parvalbumin-Interneuron Output Synapses Show Spike-Timing-Dependent Plasticity that Contributes to Auditory Map Remodeling. *Neuron* 99:720-735 e726.
- Williams SR, Mitchell SJ (2008) Direct measurement of somatic voltage clamp errors in central neurons. *Nat Neurosci* 11:790-798.
- Woodin MA, Ganguly K, Poo MM (2003) Coincident pre- and postsynaptic activity modifies GABAergic synapses by postsynaptic changes in Cl⁻ transporter activity. *Neuron* 39:807-820.

Reviewers' Comments:

Reviewer #1:

Remarks to the Author:

The authors have provided new experiments (Suppl Fig 4) and addressed my main concerns. The argumentation regarding the clamping remains a bit puzzling. I take it that the 0mV clamp experiment was in part done to isolate inhibitory currents and prevent APs. Now, the authors argue that this did not prevent involvement of T-type VGCCs since the clamp is likely to be incomplete. But then one expects this to impact the iLTP in the distal dendrites as well since the inhibitory currents will likely be very weak. Nonetheless, the authors convinced me with pharmacology experiment, which now clearly demonstrates that T-type VGCCs are engaged in this plasticity.

I have no further requests or suggestions.

ANTHONY HOLTMAAT

Reviewer #2:

Remarks to the Author:

The authors have provided reasonable responses to the criticisms.